# Learning to Schedule Learning Rate with Graph Neural Networks

**Yuanhao Xiong, Li-Cheng Lan, Xiangning Chen, Ruochen Wang, Cho-Jui Hsieh**
Department of Computer Science, UCLA
`{yhxiong, lclan, xiangning, chohsieh}@cs.ucla.edu`
`ruocwang@ucla.edu`

## Abstract

Recent decades have witnessed great development of stochastic optimization in training deep neural networks. Learning rate scheduling is one of the most important factors that influence the performance of stochastic optimizers like Adam. Traditional methods seek to find a relatively proper scheduling among a limited number of pre-defined rules and might not accommodate a particular target problem. Instead, we propose a novel Graph-Network-based Scheduler (GNS), aiming at learning a specific scheduling mechanism without restrictions to existing principles. By constructing a directed graph for the underlying neural network of the target problem, GNS encodes current dynamics with a graph message passing network and trains an agent to control the learning rate accordingly via reinforcement learning. The proposed scheduler can capture the intermediate layer information while being able to generalize to problems of varying scales. Besides, an efficient reward collection procedure is leveraged to speed up training. We evaluate our framework on benchmarking datasets, Fashion-MNIST and CIFAR10 for image classification, and GLUE for language understanding. GNS shows consistent improvement over popular baselines when training CNN and Transformer models. Moreover, GNS demonstrates great generalization to different datasets and network structures. Our code is available at `https://github.com/xyh97/GNS`.

## 1 Introduction

Stochastic optimization has achieved great successes in training deep learning models (Robbins & Monro, 1951), due to the tremendous data size and non-convexity of neural networks (Xiong et al., 2020). However, in contrast to classical and hyperparameter-free optimizers, performance of widely used stochastic algorithms such as SGD and Adam is significantly dependent on the choice of hyperparameters including batch size, learning rate and momentum coefficients (Xiong et al., 2020; Sivaprasad et al., 2020; Cooper et al., 2021; Choi et al., 2019).

Learning rate scheduling is one of the most important factors which stochastic optimizers are very sensitive to (Cooper et al., 2021; Xu et al., 2019). A fixed and static learning rate throughout the whole training is insufficient to attain an optimal model. Thus, various scheduling algorithms (Gotmare et al., 2018) including polynomial decay, cosine decay and warmup have been developed and each of them is designed in a distinct form. In practice, models trained with an appropriate scheduling mechanism can outperform those with a constant learning rate, in terms of convergence rate as well as the final performance (Senior et al., 2013; Gotmare et al., 2018; Loshchilov & Hutter, 2016).

As the learning rate scheduling is of vital importance to the model performance, how to tune it for a given stochastic optimizer effectively and automatically has been investigated intensively. Current approaches tackle this problem by assuming a particular learning rate scheduling rule (e.g., polynomial or cosine decay, warmup and restart) based on empirical studies and domain knowledge (Xu et al., 2017; Gotmare et al., 2018), and apply traditional hyperparameter optimization methods (Li et al., 2017; Snoek et al., 2012; Bergstra & Bengio, 2012) to tune the parameters of the rule. However, searching within those pre-defined principles is restricted since the best learning rate scheduling for a particular training problem may not exactly follow any existing rule.

To overcome this restriction, finding an optimal scheduler can be viewed as a learning-to-learn problem (Andrychowicz et al., 2016; Chen et al., 2020), where empirical scheduling principles are

replaced by a data-driven neural network alternative. For instance, attempts such as Xu et al. (2019) and Xu et al. (2017) model the learning rate scheduling as a sequential decision process where the learning rate is modified in every decision step, and train a policy network with reinforcement learning algorithms. However, these methods sacrifice rich state representations in order to maintain generalization. This implies that problem-dependent information, such as the architecture of neural networks to be optimized, and the state of intermediate neurons is not fully exploited. Therefore, state-of-the-art algorithms such Chen et al. (2020) only represent the current state by the statistics of final layer neurons. A straightforward way to leverage such information is to concatenate states in every layer as a super vector. However, it would lead to the explosion of the input dimension and fail to work in models with different sizes. Therefore it is nontrivial to obtain a representation of training state that is both **informative** (being able to encode rich information of architecture and intermediate neurons) and **generalizable** (can be applied to different architectures).

In this paper, we propose a Graph Network-based Scheduler (GNS) to overcome the above-mentioned challenges. Different from previous attempts, our GNS makes full use of the problem structure: since deep neural networks can be represented as directed graphs intrinsically Zhang et al. (2018), we leverage graph networks to encode states into latent features with richer information and such a design allows GNS to be easily transferred to different network architectures at the same time. Moreover, an action definition for stabling training and an efficient reward signal collection procedure are designed for our GNS framework.

Our main contributions are summarized as following:

- We develop the Graph Network-based Scheduler (GNS) to search for the best learning rate scheduling rule by representing the target network as a directed graph. Our dynamic learning rate scheduler can thoroughly capture the intermediate layer information of neural network in both image classification and language understanding tasks. Also, an efficient reward collection procedure is designed to speedup our method.
- We apply the proposed GNS to multiple training tasks. In particular, GNS achieves average scores of 85.2 and 87.8 on GLUE with RoBERTa-base and -large respectively and outperforms 84.7 and 86.9 from the best existing learning rate schedule. Note that unlike previous learning-to-learn methods that mostly are applied to toy examples such as CIFAR-10, we are the first to apply automatic learning rate scheduling to challenging tasks such as RoBERTa training and improve the performance of those realistic large models.
- We demonstrate that the GNS framework is able to generalize to problems of different datasets (from CIFAR10 to CIFAR100) and model structures (from ResNet to VGG, and from RoBERTa-base to RoBERTa-large).

## 2 RELATED WORK

**Learning rate scheduling.** This topic has been an open problem in stochastic optimization. In Almeida et al. (1998) and Baydin et al. (2017), learning rate was regarded as a trainable parameter and modified using its gradient with respect to the objective function. Sutton (1992) and Schraudolph et al. (2006) considered a meta-descent algorithm to adjust local learning rate with a meta one. In the era of deep learning, a number of adaptive optimization algorithms have been proposed, such as RMSProp (Tieleman & Hinton, 2012), AdaGrad (Duchi et al., 2011), and Adam (Kingma & Ba, 2014). They conduct parameter-wise adaptation to learning rate based on heuristics of the geometry of the traversed objective. For instance, RMSProp normalizes the learning rate with the magnitude of recent gradients while AdaGrad generates a large learning rate for infrequently-updated parameters. Complimentary to these parameter-wise adaptations, some predefined learning rate schedulers are also observed beneficial to training performance (You et al., 2019; Ge et al., 2019; Schaul et al., 2013; Senior et al., 2013). Mechanisms like polynomial (Mishra & Sarawadekar, 2019) and cosine decay (Loshchilov & Hutter, 2016) together with a warmup process (Gotmare et al., 2018) improve the convergence rate and final performance of deep learning models empirically. However, they still hold additional hyperparameters to be tuned. At the same time, these parametric schedules have limited flexibility and may not be optimized for the training dynamics of different high dimensional and non-convex optimization problems (Xu et al., 2017; 2019).

**Reinforcement learning.** The goal of reinforcement learning (RL) is to find an agent that produces an optimal policy to maximize a cumulative reward (Sutton & Barto, 2018). Recently, deep reinforcement learning with neural networks as approximation functions has shown great potentials in many

applications, such as Atari games (Gu et al., 2016; Hester et al., 2018; Schulman et al., 2017), Go (Silver et al., 2016; 2017), and training deep learning models (Zoph & Le, 2016; Cubuk et al., 2018). There are also several attempts to apply RL to learning rate controlling. For example, Daniel et al. (2016) proposed to improve the stability of training w.r.t. the initial learning rate via reinforcement learning. Li & Malik (2016), Xu et al. (2017) and Xu et al. (2019) leveraged RL agents to adjust the learning rate to reach a final performance whereas their selection of state features, reward and action leads to an inefficient training process, and is difficult to scale up to large models such as BERT (Devlin et al., 2018) and RoBERTa (Liu et al., 2019).

**Graph Neural Networks.** Graph Neural Networks (GNN) is a type of neural networks that operate on graph data and structure their computations accordingly (Battaglia et al., 2018). A graph network, or a graph-to-graph module, can be interpreted as a mapping from an input graph to an output graph with the same structure but potentially different node, edge, and graph-level representations (Battaglia et al., 2018; Sanchez-Gonzalez et al., 2020). The most important component in GNN, Message Passing Neural Networks (MPNN) (Gilmer et al., 2017), have been shown effective in learning dynamics for graph-structured settings. Generally, MPNN utilizes the graph structure to propagate information between nodes via edges, and extract latent features for downstream tasks. Variations such as GCN (Kipf & Welling, 2016) and Interaction Networks (Battaglia et al., 2016) have previously demonstrated promising results on simulation tasks in the field of physics (Battaglia et al., 2016; Sanchez-Gonzalez et al., 2020) and chemistry (Gilmer et al., 2017; You et al., 2018).

## 3 REINFORCEMENT LEARNING FOR LEARNING RATE SCHEDULING

In this section, we formulate learning rate scheduling as an RL problem, with the goal to find an optimal control policy to maximize the accumulated reward in the current environment which is defined by the target training problem.

### 3.1 REINFORCEMENT LEARNING

Reinforcement learning aims at solving a group of problems, known as Markov Decision Process (MDP) with an agent choosing optimal actions (Sutton & Barto, 2018). Typically, the MDP problem is formulated by a tuple $M = \langle \mathcal{S}, \mathcal{A}, \mathcal{P}, \mathcal{R}, \gamma \rangle$. Given the state set $\mathcal{S}$ and action set $\mathcal{A}$, the reward function $\mathcal{R}$ is a mapping of $\mathcal{S} \times \mathcal{A} \to \mathbb{R}$. The discounted return is denoted by $R_t = \sum_{l=0}^{\infty} \gamma^l r_{t+l}$ where $\gamma$ is a discount factor for future rewards. $\mathcal{P}$ is the state transition function, which specifies the probability $p(s_{t+1}|s_t, a_t)$ and is usually determined by the environment itself. To encourage more exploration, we focus on a stochastic policy $\pi(a|s)$, which maps states to action probabilities.

### 3.2 LEARNING RATE SCHEDULING AS A MDP

For most problems involved in training deep neural networks, a standard approach for the objective function minimization is stochastic gradient descent on a certain form (Robbins & Monro, 1951; Kingma & Ba, 2014; Duchi et al., 2011), which sequentially updates the parameters using gradients by $w_{k+1} = w_k - \alpha_k \cdot g_k$. Specifically, $w_k$ represents network parameters to be trained and $\alpha_k$ is the global learning rate. $g_k$ is defined by the optimizer (e.g., $g_k$ can be the SGD or Adam direction). For learning rate scheduling, we primarily consider generating a sequence for the global learning rate $\alpha_k$.

We can formulate learning rate scheduling into a sequential decision process. In each environment of a particular optimization task, states can be described by dynamics of the target network such as a combination of weight and gradient values (Xu et al., 2019; Daniel et al., 2016). Actions can be any operation that might change the learning rate. As to rewards, since the ultimate goal is to achieve higher final performance, metrics like loss or accuracy can serve as a signal to guide the agent training. Besides, as training a deep learning model usually requires a large number of iterations, rather than a single optimization step, we execute an action every $K$ steps to avoid instability from too frequent learning rate modifications. By training such an agent modeled as a policy network, we can obtain a learning rate scheduler to adjust learning rate dynamically.

## 4 METHOD

### 4.1 STATE ENCODING WITH GRAPH NEURAL NETWORKS

The selection of features to represent the environment is of vital significance. Previous work such as (Xu et al., 2017) and (Xu et al., 2019) aims to learn a generalized agent that can be applicable to any

network architecture, so they were only able to utilize statistics of the final layer to describe the state. It might work for small scale models, but when dealing with large ones like RoBERTa (Liu et al., 2019), solely using information of the last layer is insufficient to describe the environment.

A straightforward way to obtain a more thorough representation is to utilize dynamics of all layers in a neural network and concatenate them into one super vector. However, such a representation cannot guarantee promising results as shown in Section 5.6 of ablation study. The problem is that naively concatenating all features increases the input dimension and does not explore the correlation between layers. For instance, dynamics between nearby layers should have a higher correlation than layers that are far away. This concatenation also loses transferrablity to different network structures.

Therefore, to maintain a rich state description and generalization in the meanwhile, we construct a computational graph to depict the state. We represent a network structure as a directed graph $G = (\mathcal{E}, \mathcal{V})$. Each node $v$ denotes an operator $f_v(\cdot; w_v)$ (convolution, linear, etc.) parameterized by $w_v$ which generates an output activation tensor. An edge $e_{uv} \in \mathcal{E}$ represents the computation flow from node $u$ to node $v$. The output for node $v$ is computed by applying its associated computational operator taking all source nodes as the input. Examples of the graph structure of a neural network are presented in Figure 1(a) and (b). For each node $v$ with trainable parameters $w_v$, the raw feature $x_v$ includes the mean and variance of weights, biases, and absolute gradient values of $w_v$, as well as global dynamics of the previous learning rate and the average training loss in the decision step. In the end, the environment can be represented by a graph with a feature matrix $\mathbf{X} \in \mathbb{R}^{N \times F}$ and the adjacency matrix $\mathbf{A} \in \mathbb{R}^{N \times N}$, where $N$ is the number of nodes and $F$ is the feature dimension.

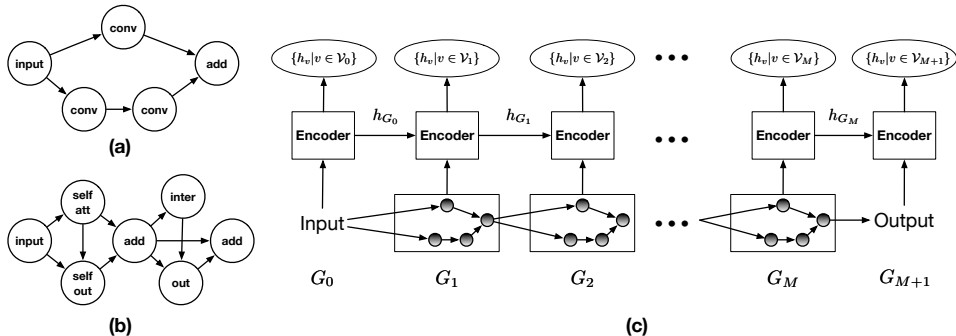

**(a)**

**(b)**

**(c)**

Figure 1: (a) A repeated block in ResNet models. (b) A repeated block in Transformer models. (c) Illustration of the hierarchical GCN message passing.

Given an input architecture, we describe the state as a graph $G$ with the feature matrix $\mathbf{X}$ and the adjacency matrix $\mathbf{A}$. Then we can feed $G$ into an encoder $E$, modeled as a graph convolutional neural network (GCN) (Kipf & Welling, 2016), one of the most effective techniques to achieve message propagation. More concretely, hidden features can be extracted by

$$\mathbf{H}^{l+1} = E(\mathbf{H}^l, \mathbf{A}) = \sigma(\tilde{\mathbf{A}}\mathbf{H}^l\mathbf{W}^l), \tag{1}$$

where $\mathbf{H}^0$ is initialized as $\mathbf{X}$, $\tilde{\mathbf{A}}$ is the normalized adjacency matrix with a self-loop in each node, and $\mathbf{W}^l$ is the parameters of the $l$-th layer of GCN.

However, a deep learning model usually adopts hundreds of operations and results in a large computation graph which is hard to process. We notice that repeated blocks are stacked in commonly-used structures such as ResNet (He et al., 2016) and Transformer (Devlin et al., 2018; Vaswani et al., 2017) to boost the expressiveness of the network (Zhang et al., 2018; Zoph et al., 2018), shown in Figure 1(a) and (b). Instead of passing messages through the whole graph directly, we separate the graph into a number of blocks as $\{G_i = (\mathcal{V}_i, \mathcal{E}_i)\}_{i=0}^{M+1}$ and each block has a group of nodes. $M$ is the number of repeated blocks, and $G_0$ stands for the input block (e.g., embedding layers) while $G_{M+1}$ represents the output block (e.g., classification layers). We can then use a shared encoder across different blocks as shown in Figure 1(c) to conduct information propagation. This is implemented by a hierarchical GCN as follows: **(i)** compute latent features of nodes in each block by Eq. 1; **(ii)** summarize the $i$-th block as $h_{G_i} = \text{AGG}(\{h_v | v \in \mathcal{V}_i\})$; **(iii)** then use $h_{G_i}$ as the input node feature for the next block: **(iv)** Finally, to obtain the graph level information to describe the whole network, we just take the aggregation over each node by $h_G = \text{AGG}(\{h_v | v \in \mathcal{V}\})$. Here, we use the mean function as aggregation function to allow generalization to different structures.

Compared with a concatenated state with dynamics in all nodes, representing the environment as a graph and then encoding it via a hierarchical GCN have two primary advantages: (1) GCN can capture the correlation between neighboring nodes and generate a meaningful latent feature for action prediction; (2) a shared encoder can learn message passing within one block and between blocks in the meanwhile and is not constrained to a specific graph structure. In detail, the dimension of a concatenated representation varies between different network architectures, for example, RoBERTa-base and -large models. A scheduler learned from such concatenated states fails to work in another environment due to the input mismatch. On the contrary, by stacking the same encoder along the depth dimension, we can transfer our graph-network-based scheduler trained on one network to the new environment of another different network directly or with slight fine-tuning.

## 4.2 Environment design

We discuss the remaining components of the environment of the target problem involved in neural network optimization in this part besides the state. An effective set of reward functions and actions is defined to facilitate an efficient and generalizable agent.

**Action.** We adopt the action as scaling the previous learning rate by a factor (Xu et al., 2019) considering the wide range of $\alpha$. Through that operation, the learning rate can be changed stably and consistently. More concretely, after each action execution, $\alpha_{t+1} = \alpha_t \cdot (1 + a_t)$.

**State transition.** As mentioned in Section 3.2, once we modify the learning rate to $\alpha_{t+1}$, we execute the update rule of gradient descent for $K$ batches to get the next state $s_{t+1}$.

**Reward design.** With the goal of the agent to achieve higher final performance, there are a series of metrics that can be leveraged, such as validation loss and accuracy. In Xu et al. (2019), validation loss at each decision step serves as an immediate reward. However, loss might not be an ideal indicator for task specific metrics which we are ultimately concerned with, like accuracy, F1 score and correlation (Venter et al., 2021). Thus, we choose to use such task-related metrics directly instead of the loss. On the other hand, since defining an intermediate reward can utilize training information thoroughly and provide direct feedback (Xu et al., 2019), we take the performance difference between two steps as the reward signal:

$$r_t = P_{t+1} - P_t, \tag{2}$$

where $P_t$ represents the validation result under the given metric at the $t$-th decision step. Whereas, computing a reward in Eq. 2 indicates that we have to conduct a validation procedure every decision step, which is a non-trivial burden. In order to make training more efficient, we turn to reduce the frequency of validation with a probability $p$, i.e., the validation is only carried out if the number randomly sampled from $\text{Uni}(0, 1)$ is less than $p$. Specifically, suppose under our principle, we calculate the validation metric $P_{t_1}$ and $P_{t_2}$ at the decision step $t_1$ and $t_2$ respectively. The reward signal for each step can be derived by linearly distributing the difference as follows:

$$r'_t = \frac{P_{t_2} - P_{t_1}}{t_2 - t_1}, \forall t \in [t_1, t_2). \tag{3}$$

## 4.3 Graph Network-based Scheduler

With illustration of the environment design, we can then move to the general framework of our Graph Network-based Scheduler (GNS). As shown in Figure 2, at each decision step, GNS takes the observation $s_t$ as input, encodes $s_t$ into $h_t$, sample an action $a_t$ based on the policy network, evaluates the value of $s_t$, then execute the action to render the next state $s_{t+1}$.

Figure 2: The transition from $s_t$ to $s_{t+1}$.

In the procedure of sampling and evaluation, we use the hidden feature $h_t$ to evaluate the current state and predict the action. Denote $m_s$ and $m_f$ as two different MLPs. Specifically, the state value can be computed by $V(s_t) = m_s(h_t)$. Then we calculate the mean of the action and the policy $\pi$ samples $a_t$ from the corresponding normal distribution in Eq. 4 with a fixed deviation $\sigma$:

$$a_t \sim \mathcal{N}(\mu_t, \sigma^2), \; \mu_t = 0.1 \cdot \tanh(m_f(h_t)). \tag{4}$$

By Eq. 4, we can constrain that the scale factor $(1 + a_t)$ is within the range of $[0.9, 1.1]$ at most of the times in order to train the target network stably (Xu et al., 2019).

As to the agent training, we first parameterize the value network $V$ (critic), and policy network $\pi$ (actor) by $\phi, \varphi$ respectively for clarity. Each environment of the target problem has a fixed time horizon $T$, i.e., the number of total training steps, which is called as one episode. We adopt Proximal Policy Optimization (PPO) (Schulman et al., 2017) in the actor-critic style to optimize our scheduling policy, which is the state of the art on-policy RL methods in many continuous control tasks (Achiam, 2018; Berner et al., 2019). On-policy RL methods only use the experience generated by current policy. Therefore, the training will be more stable due to the training data distribution. The key objective function of the policy network is defined as

$$L^{\text{clip}}(\varphi) = \mathbb{E}_t \left[ \min(\rho_t(\varphi)\hat{A}_t, \text{clip}(\rho_t(\varphi), 1 - \epsilon, 1 + \epsilon)\hat{A}_t) \right], \rho_t(\varphi) = \frac{\pi_\varphi(a_t|s_t)}{\pi_{\varphi_{\text{old}}}(a_t|s_t)}, \quad (5)$$

where $\rho_t(\varphi)$ is the probability ratio and $\hat{A}_t$ is the estimated advantage value. Besides, a learned state-value function $V_\phi(\cdot)$ is leveraged to reduce variance in advantage estimators and can be trained by minimizing the following squared-error loss:

$$L(\phi) = \mathbb{E}_t[(V_\phi(s_t) - V_t^{\text{targ}})^2], \quad (6)$$

where $V_t^{\text{targ}} = r_t + \gamma V(s_{t+1})$. During the training phase, we run the agent in the environment for one episode, and then update the value and policy network at the end. We reset environment at the beginning of each episode and repeat the process for multiple episodes. The detailed algorithm is presented in Appendix A. When applying the scheduler, we first sample a number of initial learning rates, run each setting for $T_{\text{pre}}$ steps, compute corresponding $V(s_{T_{\text{pre}}})$, and choose the learning rate with the largest value to complete the training.

## 5 EXPERIMENTAL RESULTS

To validate the effectiveness of GNS, we evaluate our method on various tasks in image classification and language understanding, and compare it with popular learning rate scheduling rules. We further investigate the generalization of GNS on different transfer tasks. In addition, we conduct an ablation study to analyze the state representation and reward collection.

### 5.1 EXPERIMENTAL SETTINGS

In this section, we present our experimental settings. Further details can be found in Appendix B.

**Image classification.** We consider two benchmark datasets in image classification, Fashion-MNIST (Xiao et al., 2017) and CIFAR10 (Krizhevsky et al., 2014). These two datasets are first split into the standard training and test sets. Then we randomly sample 10k images for each dataset from the training set to construct a validation set. A ResNet-18 (He et al., 2016) model is trained on Fashion-MNIST while a ResNet-34 (He et al., 2016) is trained on CIFAR10. We use Adam (Kingma & Ba, 2014) with the batch size of 128 for 200 epochs to train these two image classification tasks.

**Language understanding.** For language understanding, we conduct experiments on GLUE (Wang et al., 2019), a benchmark consisting of eight sentence- or sentence-pair tasks. They are divided into training, validation and test sets and we have no access to ground truth labels of test sets. On each dataset, we fine-tune a pre-trained RoBERTa model (Liu et al., 2019), use it to generate predictions on the test set, and obtain the corresponding performance through the official GLUE platform. Both RoBERTa-base and -large models are evaluated. No ensembling techniques are adopted for purely validating the effectiveness of our proposed scheduler. The AdamW (Loshchilov & Hutter, 2017) optimizer is adopted to train RoBERTa models. Details of other hyperparameters like batch size and episode length for each task are provided in Appendix B.

**Baselines.** Three hand-designed scheduling mechanisms (constant, polynomial decay, cosine decay), one gradient-based method Hypergradient (Baydin et al., 2017), and one learning-based method, SRLS (Xu et al., 2019) are compared with our GNS framework as baselines. For traditional schedulers

with the warmup process, we use the following equation to summarize them:

$$\alpha_t = \begin{cases} \frac{t_{\text{cur}}}{T_{\text{warm}}} \alpha_{\max}, & \text{warmup;} \\ (\alpha_{\max} - \alpha_{\min}) \left(1 - \frac{t_{\text{cur}} - T_{\text{warm}}}{T_{\text{decay}} - T_{\text{warm}}}\right)^p + \alpha_{\min}, & \text{polynomial;} \\ \left[\frac{1}{2}(\alpha_{\max} - \alpha_{\min})\left(1 + \cos(\frac{t_{\text{cur}} - T_{\text{warm}}}{T_{\text{decay}} - T_{\text{warm}}}\pi)\right) + \alpha_{\min}\right], & \text{cosine;} \\ \alpha_{\min}, & \text{constant.} \end{cases} \tag{7}$$

In detail, $\alpha_{\max}$ and $\alpha_{\min} = \mu\alpha_{\max}$ are the maximum and minimum learning rate respectively during the training. $T_{\text{total}}$ is the number of total training steps, $T_{\text{warm}} = \eta_1 T_{\text{total}}$ is the number of warmup steps and $T_{\text{decay}} = \eta_2 T_{\text{total}}$ is the number of decay steps. We tune all hyperparameter combinations within the search space in Appendix B and select the configuration with the best validation performance. Then we run the experiment 5 times and report the average test performance. For Hypergraident and SRLS, we follow training details in Baydin et al. (2017) and Xu et al. (2019) respectively.

**GNS implementation.** To encode raw observations into latent features, we use a 2-layer GCN (Kipf & Welling, 2016) with 64 dimensional node embeddings in all hidden layers, and apply batch normalization for each layer. After a shared encoder, each of the value network and policy network has an MLP with a hidden layer of size 32. Our GNS is trained with PPO (Schulman et al., 2017) for 30 episodes for ResNet models and RoBERTa-base while 10 episodes for RoBERTa-large by fine-tuning the agent learned from RoBERTa-base, by an Adam optimizer of the learning rate 0.005 and 0.001 for the critic and actor respectively.

## 5.2 RESULTS ON IMAGE CLASSIFICATION

In this section, we present experimental results of image classification tasks. Test accuracy obtained by different schedulers are shown in Table 1. It can be observed that the constant learning rate scheduling is insufficient to lead to a satisfactory result and is beaten by all other methods. Although hand-designed schedulers like polynomial and cosine decay and the gradient-based method Hypergradient can make an improvement over a constant learning rate, learning-based methods can still outperform them. In particular, GNS achieves much better accuracy on both Fashion-MNIST and CIFAR10 than SRLS, due to a richer state representation from the utilization of the model's graph structure.

Table 1: Test accuracy on Fashion-MNIST and CIFAR10 of different schedulers

| Scheduler | Fashion-MNIST | CIFAR10 |
|---|---|---|
| Constant | $93.0 \pm 0.2$ | $93.0 \pm 0.4$ |
| Polynomial | $93.4 \pm 0.1$ | $93.5 \pm 0.3$ |
| Cosine | $93.4 \pm 0.1$ | $93.6 \pm 0.4$ |
| Hypergradient | $93.5 \pm 0.2$ | $93.7 \pm 0.6$ |
| SRLS | $93.6 \pm 0.3$ | $93.6 \pm 0.6$ |
| GNS | $\mathbf{94.6 \pm 0.2}$ | $\mathbf{94.3 \pm 0.5}$ |

## 5.3 RESULTS ON LANGUAGE UNDERSTANDING

Apart from image classification, we also focus on language understanding tasks on eight GLUE benchmark datasets. Two models with different sizes, RoBERTa-base and RoBERTa-large, are evaluated and detailed results are reported in Table 2. As expected, adjusting learning rate rather than keeping a constant one contributes to the test performance. In addition, a consistent performance boost can be observed in GNS over all eight datasets for both base and large models. Specifically, it achieves a score of 85.2 and 87.8 on average in RoBERTa-base and -large model respectively, while the best baseline in each scenario only reaches 84.5 and 86.9. It should be noticed that SRLS performs poorly and even cannot surpass the constant scheduling. It is likely that the simple state design of SRLS (only using information in the last layer) constrains its ability to adjust the learning rate dynamically, especially for large models like RoBERTa. Thus, SRLS fails to generate a good scheduling to improve the performance.

We plot learning rate scheduling trajectories for three tasks in Figure 3: MRPC, RTE, and STS-B. It can be observed that different mechanisms are deployed for distinct tasks. The learning rate is

Table 2: Test performance of RoBERTa on GLUE benchmarking of different schedulers.

| Scheduler | CoLA | SST-2 | MRPC | QQP | STS-B | MNLI | QNLI | RTE | Avg. |
|---|---|---|---|---|---|---|---|---|---|
| *Base model:* | | | | | | | | | |
| Constant | $60.5 \pm 0.4$ | $95.1 \pm 0.2$ | $86.9 \pm 0.6$ | $89.0 \pm 0.0$ | $88.6 \pm 0.2$ | $87.0 \pm 0.1$ | $92.9 \pm 0.1$ | $73.0 \pm 0.3$ | 84.1 |
| Polynomial | $61.2 \pm 0.3$ | $95.3 \pm 0.3$ | $87.2 \pm 0.4$ | $89.2 \pm 0.1$ | $89.3 \pm 0.3$ | $87.3 \pm 0.2$ | $93.0 \pm 0.1$ | $73.1 \pm 0.4$ | 84.4 |
| Cosine | $61.2 \pm 0.3$ | $95.4 \pm 0.3$ | $87.2 \pm 0.5$ | $89.2 \pm 0.1$ | $89.4 \pm 0.1$ | $87.2 \pm 0.1$ | $93.0 \pm 0.2$ | $73.5 \pm 0.3$ | 84.5 |
| Hypergradient | $61.0 \pm 0.4$ | $95.3 \pm 0.2$ | $87.0 \pm 0.4$ | $89.1 \pm 0.2$ | $89.4 \pm 0.2$ | $87.1 \pm 0.1$ | $93.0 \pm 0.1$ | $73.4 \pm 0.5$ | 84.4 |
| SRLS | $58.4 \pm 1.2$ | $95.3 \pm 0.3$ | $86.8 \pm 0.5$ | $88.4 \pm 0.1$ | $89.4 \pm 0.3$ | $86.7 \pm 0.2$ | $92.3 \pm 0.2$ | $70.8 \pm 0.7$ | 83.5 |
| GNS | $\mathbf{63.3 \pm 0.5}$ | $\mathbf{95.8 \pm 0.2}$ | $\mathbf{87.7 \pm 0.5}$ | $\mathbf{89.6 \pm 0.0}$ | $\mathbf{89.7 \pm 0.1}$ | $\mathbf{87.7 \pm 0.2}$ | $\mathbf{93.2 \pm 0.0}$ | $\mathbf{74.4 \pm 0.3}$ | **85.2** |
| *Large model:* | | | | | | | | | |
| Constant | $63.1 \pm 0.7$ | $95.8 \pm 0.2$ | $87.7 \pm 0.7$ | $89.2 \pm 0.0$ | $90.1 \pm 0.3$ | $90.0 \pm 0.2$ | $94.0 \pm 0.2$ | $80.6 \pm 0.2$ | 86.3 |
| Polynomial | $64.2 \pm 0.8$ | $95.9 \pm 0.4$ | $88.4 \pm 0.7$ | $89.2 \pm 0.1$ | $91.1 \pm 0.2$ | $90.0 \pm 0.0$ | $94.5 \pm 0.1$ | $82.0 \pm 0.2$ | 86.9 |
| Cosine | $63.3 \pm 0.6$ | $95.8 \pm 0.3$ | $88.5 \pm 0.6$ | $89.3 \pm 0.0$ | $91.2 \pm 0.2$ | $89.9 \pm 0.2$ | $94.3 \pm 0.2$ | $81.0 \pm 0.3$ | 86.7 |
| Hypergradient | $63.3 \pm 0.7$ | $95.7 \pm 0.4$ | $88.2 \pm 0.8$ | $89.2 \pm 0.1$ | $91.0 \pm 0.3$ | $89.9 \pm 0.3$ | $94.4 \pm 0.2$ | $81.4 \pm 0.4$ | 86.6 |
| SRLS | $62.3 \pm 0.5$ | $95.9 \pm 0.3$ | $87.5 \pm 1.2$ | $88.2 \pm 0.2$ | $91.2 \pm 0.4$ | $88.8 \pm 0.1$ | $93.4 \pm 0.2$ | $79.8 \pm 0.3$ | 85.9 |
| GNS | $\mathbf{65.8 \pm 0.7}$ | $\mathbf{96.7 \pm 0.1}$ | $\mathbf{89.0 \pm 0.9}$ | $\mathbf{89.7 \pm 0.1}$ | $\mathbf{92.1 \pm 0.2}$ | $\mathbf{90.4 \pm 0.1}$ | $\mathbf{94.7 \pm 0.1}$ | $\mathbf{84.0 \pm 0.2}$ | **87.8** |

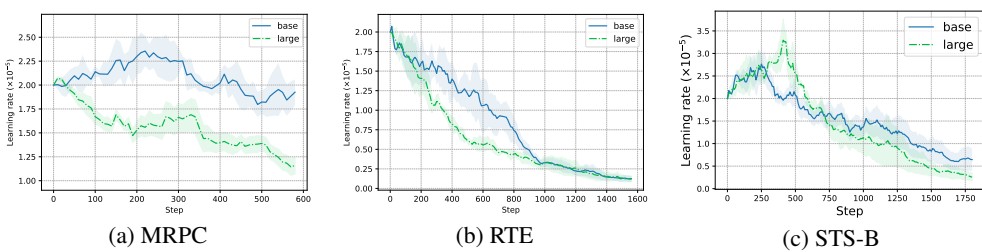

(a) MRPC        (b) RTE        (c) STS-B

Figure 3: Learning rate trajectories generated by GNS for three tasks.

generally adjusted in a similar trend for both RoBERTa-base and large models within a task. Specifically, we can see that the learning rate for MRPC is not modified aggressively. For RTE, a decayed scheduling is learned while there is a warmup-like procedure followed by a decreasing tendency for STS-B. Besides, it should be emphasized that the learning rate does not change monotonically, and there are ups and downs during the training to react to the environment observations dynamically.

## 5.4 GENERALIZATION OF GNS

Due to the utilization of hierarchical GCN as the learning rate scheduler described in Section 4.3, our GNS can be transferred to different but similar graphs, e.g., ResNet and VGG (Conneau et al., 2016) structures on CIFAR10. Thus, we conduct an experiment to evaluate the generalization of the learned scheduler. We select the policy network trained under the environment of ResNet-34, and apply it directly to the task with VGG-16. For all other baselines, we simply adopt the best hyperparameter configuration of ResNet-34 to train the VGG-16 network. Results are shown in the second column of Table 3. We can see that GNS can still achieve the best performance on the VGG model while the hyperparameter settings of other methods are not suitable for the VGG model. It indicates that GNS is able to learn the fundamental rule to conduct message passing between nodes and is generalizable to various graphs. We also evaluate the generalization of GNS on language tasks from RoBERTa-base model to large model in Appendix E. The excellent transferred performance also explains why only a few episodes are needed when fine-tuning the agent under RoBERTa-large models. In addition, GNS can be utilized to the same network structure but on different datasets.

Table 3: Transferred performance of GNS.

| Scheduler | ResNet $\rightarrow$ VGG | CIFAR10 $\rightarrow$ CIFAR100 | CIFAR10 $\rightarrow$ ImageNet |
|---|---|---|---|
| Constant | $91.4 \pm 0.3$ | $69.3 \pm 0.3$ | 68.9 |
| Polynomial | $91.8 \pm 0.3$ | $70.5 \pm 0.4$ | 70.3 |
| Cosine | $91.7 \pm 0.4$ | $71.2 \pm 0.2$ | 70.5 |
| Hypergradient | $91.5 \pm 0.5$ | $70.9 \pm 0.3$ | 70.0 |
| SRLS | $91.7 \pm 0.3$ | $71.0 \pm 0.6$ | 71.6 |
| GNS | $\mathbf{92.5 \pm 0.4}$ | $\mathbf{72.3 \pm 0.3}$ | **73.1** |

Specifically, we apply the graph scheduler trained on ResNet-34 for CIFAR10 to training a ResNet-34 network for CIFAR100 and ImageNet. It can be observed in Table 3 that GNS achieves 72.3% on test

accuracy and outperforms 71.2% of the best baseline scheduler with a cosine decay on CIFAR100. On ImageNet, transferred performance of GNS is 73.1% and is still better than the best baseline of 71.6%. Such a performance further validates that GNS is generalizable to different datasets.

## 5.5 RUNNING TIME COMPARISON

Compared with SRLS in which validation is required in every decision step, GNS can reduce a considerable amount of time with the validation probability $p = 0.2$. We consider training RoBERTa-base in this section. For instance, when running on MRPC for 5 epochs, we need to make 58 decisions with the number of network updates $K = 10$. The average time of one episode with one NVIDIA 1080Ti GPU of SRLS is 405s while GNS only takes 259s, which decreases the original cost by 30%.

For hand-designed rules, there are 40 hyperparameter settings shown in Appendix B to be considered for RoBERTa fine-tuning while it takes 30 episodes to train the learned scheduler. The time for running one setting is 223s on average. Thus, a thorough grid search among these 40 settings consumes approximately 2.5 hours. On the contrary, GNS takes 2.2 hours and even achieves better results. Furthermore, fine-tuning the scheduler on RoBERTa-large from the base model only requires 10 episodes and costs much less time than grid search in all configurations.

## 5.6 ABLATION STUDY

In this part, an ablation study is carried out to corroborate that the design of state representations and reward collection play an important role in the GNS framework.

Table 4: Test performance of three scheduler variants for RoBERTa-base.

| Scheduler | CoLA | SST-2 | MRPC | QQP | STS-B | MNLI | QNLI | RTE | Avg. |
|-----------|------|-------|------|-----|-------|------|------|-----|------|
| SRLS | $58.4 \pm 1.2$ | $95.3 \pm 0.3$ | $86.8 \pm 0.5$ | $88.4 \pm 0.1$ | $89.4 \pm 0.3$ | $86.7 \pm 0.2$ | $92.3 \pm 0.2$ | $70.8 \pm 0.7$ | 83.5 |
| CSS | $60.0 \pm 0.5$ | $94.9 \pm 0.1$ | $87.1 \pm 0.3$ | $89.2 \pm 0.2$ | $88.7 \pm 0.3$ | $86.4 \pm 0.1$ | $92.3 \pm 0.2$ | $72.2 \pm 0.5$ | 83.8 |
| FRS | $\mathbf{63.5 \pm 0.3}$ | $\mathbf{95.9 \pm 0.3}$ | $\mathbf{87.7 \pm 0.4}$ | $89.4 \pm 0.0$ | $89.4 \pm 0.2$ | $87.6 \pm 0.2$ | $\mathbf{93.2 \pm 0.1}$ | $\mathbf{74.6 \pm 0.4}$ | 85.2 |
| GNS | $\mathbf{63.3 \pm 0.5}$ | $\mathbf{95.8 \pm 0.2}$ | $\mathbf{87.7 \pm 0.5}$ | $\mathbf{89.6 \pm 0.0}$ | $\mathbf{89.7 \pm 0.1}$ | $\mathbf{87.7 \pm 0.2}$ | $\mathbf{93.2 \pm 0.0}$ | $\mathbf{74.4 \pm 0.3}$ | 85.2 |

**State representations with graph networks.** To show advantages in encoding the state by a graph neural network, we compare our GNS with a variant called concatenated-state scheduler (CSS), where features of all nodes in the state graph are concatenated to form a super representation and then fed into subsequent neural networks for controlling the learning rate. Compared with the basline SRLS that only uses the final layer state, CSS uses all layers but without using GCN to exploit correlation between layers and cannot be applied to different target models. We can observe from results in Table 4 that using a concatenated state slightly outperforms SRLS but is still insufficient to render as a competitive scheduler as GNS, which shows the importance of encoding by graph neural networks.

**Reward collection with a probability.** We also compare the difference between collecting a reward signal at every decision step (the variant is denoted as full-reward scheduler, FRS) and collecting it with a probability. We report the evaluation metric on the test set in Table 4. We can see that an instant reward cannot necessarily improve the final performance of the learned scheduler. It is likely that the performance does not improve monotonically and typically oscillates during the training. For example, in two consecutive decision steps, the latter one gives us a worse validation result, leading to a negative reward computed by Eq. 2 and indicating that the action is not good. However, this information might be misleading and impede agent training. Thus, to alleviate such a phenomenon, we design a stochastic reward collection procedure, expecting to obtain more useful signals.

## 6 CONCLUSION

This paper proposes a novel graph-network-based scheduler to control the learning rate dynamically by reinforcement learning. We are the first to take into consideration the structure of the target optimization problem with neural networks, and hereby leverage message passing via the graph to encode the state for predicting the action. Comprehensive experiments have shown that GNS can consistently achieve performance boosts on benchmark datasets in both image classification and language understanding. However, in this paper we only focus on problems with a moderate time horizon. How to efficiently train a learning rate scheduler for a problem with hundreds of thousands optimization steps such as BERT pre-training with techniques like early stopping and successive halving can be a potential future direction.

ETHICS STATEMENT

To the best of our knowledge, there are no ethical issues with this paper.

REPRODUCIBILITY STATEMENT

To ensure reproducibility of our experiments, we have provided comprehensive details in the main paper as well as in the appendix. Specifically, we introduce the experimental settings in Section 5.1 including selected datasets and implementation details of all compared methods. Hyperparameter configurations and tuning details are presented in Appendix B. Moreover, we have also uploaded our source codes as supplementary materials for reference.

ACKNOWLEDGEMENT

This work is supported by NSF under IIS-2008173, IIS-2048280 and by Army Research Laboratory under agreement number W911NF-20-2-0158.

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

## A  GNS ALGORITHM

---

**Algorithm 1** Graph Network-based Scheduler.

---

**Input:** Value network parameterized by $\phi$, action network parameterized by $\varphi$, # updates $T$, decision interval $K$, prior learning rate distribution $\mathcal{D}_\alpha$

1: **for** episode $= 1, 2, \ldots$ **do**
2:     $t \leftarrow 1$
3:     Sample an initial $\alpha_t$ from the prior distribution $\mathcal{D}_\alpha$
4:     **for** $i = 1$ **to** $T$ **do**
5:         Conduct one optimization step by $w_{t+1} = w_t - \alpha_t \cdot g_t$ with $\alpha_t$
6:         **if** $i \bmod K == 0$ **then**
7:             Obtain the graph state representation $s_t$
8:             Sample an action $a_t$ and modify the learning rate by $\alpha_{t+1} = \alpha_t \cdot (1 + a_t)$
9:             Conduct validation as described in Section 4.2
10:             $t \leftarrow t + 1$
11:         **end if**
12:     **end for**
13:     Calculate rewards by Eq. (3) and organize the trajectory as $\mathcal{T} = \{(s_t, a_t, r_t)\}_{t=1}^{T//K}$
14:     Update $\phi$ and $\varphi$ using $\mathcal{T}$ based on PPO algorithm
15: **end for**

---

# B ADDITIONAL IMPLEMENTATION DETAILS

In this section, additional implementation details are provided for reproducing experimental results.

## B.1 SEARCH SPACE FOR HAND-DESIGNED SCHEDULERS

As mentioned in Section 4.1, we need to conduct hyperparameter optimization on hand-designed learning rate schedulers. The search spaces for image classification and language understanding tasks are presented in Table 5 and 6 respectively, which are selected based on previous work Choi et al. (2019); Liu et al. (2019); Devlin et al. (2018).

Table 5: Search space for image classification

| Hyperparameter | Tuning range |
|---|---|
| $\alpha_{max}$ | $[10^{-1}, 10^{-2}, 10^{-3}]$ |
| $\mu$ | $[10^{-1}, 10^{-2}, 10^{-3}]$ |
| $\eta_2$ | $[0.5, 1.0]$ |
| $p$ | $[1, 2]$ |

Table 6: Search space for language understanding

| Hyperparameter | Tuning range |
|---|---|
| $\alpha_{max}$ | $[1 \times 10^{-5}, 2 \times 10^{-5}]$ |
| $\eta_1$ | $[0, 0.025, 0.05, 0.075, 0.1]$ |
| $p$ | $[1, 2]$ |

## B.2 EXPERIMENTAL SETTINGS ON IMAGE CLASSIFICATION

We present a detailed hyperparameter configuration for two image classification tasks in Table 7.

Table 7: Hyperparameter configuration for GLUE benchmarking datasets.

| | Fashion MNIST | CIFAR10 |
|---|---|---|
| batch size | 128 | 128 |
| # updates $T$ | 78200 | 62600 |
| decision interval $K$ | 500 | 500 |
| Metric | Acc | Acc |

## B.3 EXPERIMENTAL SETTINGS ON GLUE

We further elaborate on details of GLUE benchmarking. All RoBERTa models in this paper are implemented by Hugging Face[1] Wolf et al. (2020) and pre-trained models are obtained from the corresponding model hub[2]. We use suggested values of batch size and the number of updates in Liu et al. (2019) for all eight tasks, as shown in Table 8. In addition, the decision interval and the metric for reward computation of each dataset are provided. All tasks are optimized by AdamW with the weight decay of $0.1$.

---

[1]https://github.com/huggingface/transformers
[2]https://huggingface.co/roberta-base and https://huggingface.co/roberta-large

Table 8: Hyperparameter configuration for GLUE benchmarking datasets.

|                      | CoLA        | SST-2 | MRPC | QQP   | STS-B       | MNLI  | QNLI  | RTE  |
|----------------------|-------------|-------|------|-------|-------------|-------|-------|------|
| batch size           | 16          | 32    | 32   | 32    | 16          | 32    | 32    | 16   |
| # updates $T$        | 2680        | 10525 | 575  | 56855 | 1800        | 61360 | 16370 | 1560 |
| decision interval $K$| 10          | 100   | 10   | 1000  | 10          | 1000  | 100   | 10   |
| Metric               | Corr[1]     | Acc   | Acc  | Acc   | Corr[2]     | Acc   | Acc   | Acc  |

[1] Mathew's correlation.

[2] Pearson correlation.

### B.4 HYPERGRADIENT

We present the details of Hypergradient in Algorithm 2. We still have two tunable hyperparameters for Hypergradient, $\alpha_0$ and $\beta$. The search space for image classification is: $\alpha_0 \in [10^{-1}, 10^{-2}, 10^{-3}]$ and $\beta \in [10^{-7}, 10^{-8}, 10^{-9}]$. The search space for language understanding is: $\alpha_0 \in [1 \times 10^{-5}, 2 \times 10^{-5}]$ and $\beta \in [10^{-7}, 10^{-8}, 10^{-9}]$.

---

**Algorithm 2** Adam with Hypergradient descent (Adam-HD)

---

**Require:** : $\alpha_0$: initial learning rate
**Require:** : $\beta$: hypergradient learning rate
**Require:** $\beta_1, \beta_2 \in [0, 1)$: decay rates for Adam
$\quad t, m_0, v_0, \nabla_\alpha u_0 \leftarrow 0, 0, 0, 0$                    ▷ Initialization
$\quad$ **Update rule:**
$\quad m_t \leftarrow \beta_1\, m_{t-1} + (1 - \beta_1)\, g_t$          ▷ 1st mom. estimate
$\quad v_t \leftarrow \beta_2\, v_{t-1} + (1 - \beta_2)\, g_t^2$          ▷ 2nd mom. estimate
$\quad \widehat{m}_t \leftarrow m_t / (1 - \beta_1^t)$              ▷ Bias correction
$\quad \widehat{v}_t \leftarrow v_t / (1 - \beta_2^t)$               ▷ Bias correction
$\quad u_t \leftarrow -\alpha_t\, \widehat{m}_t / (\sqrt{\widehat{v}_t} + \epsilon)$      ▷ Parameter update
$\quad \nabla_\alpha u_t \leftarrow -\widehat{m}_t / (\sqrt{\widehat{v}_t} + \epsilon)$

---

### B.5 FURTHER COMPARISON OF SRLS AND GNS

Due to the property of shared parameters in graph neural networks, our GNS does not introduce many extra parameters and it takes similar time to train GNS compared with training the original policy network in SRLS. On the other hand, although the efficient reward collection can be applied to SRLS as well, there is still an obvious performance gap between GNS and SRLS, as shown in Table 9.

Table 9: Test accuracy of SRLS and GNS under reward collection with $p = 0.2$.

| Scheduler | Fashion-MNIST   | CIFAR10        |
|-----------|-----------------|----------------|
| SRLS      | $93.5 \pm 0.4$  | $93.4 \pm 0.3$ |
| GNS       | $\mathbf{94.6 \pm 0.2}$ | $\mathbf{94.3 \pm 0.5}$ |

## C GRAPH REPRESENTATION OF ROBERTA

We present a complete graph representation of RoBERTa-base model in Figure 4 for clarity. There are 12 repeated blocks in total in RoBERTa-base.

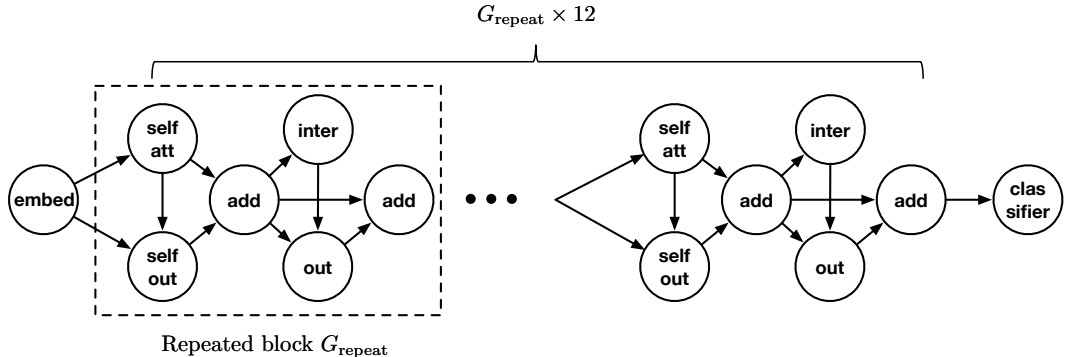

Figure 4: A graph representation of RoBERTa-base model.

# D    DETAILS OF PPO

As mentioned in Section 3.5, we adopt Proximal Policy Optimization (PPO) Schulman et al. (2017) in the actor-critic style to train our scheduling policy. The key objective function of the policy network is defined as

$$L^{\text{clip}}(\varphi) = \mathbb{E}_t \left[ \min(\rho_t(\varphi)\hat{A}_t, \text{clip}(\rho_t(\varphi), 1 - \epsilon, 1 + \epsilon)\hat{A}_t) \right], \rho_t(\varphi) = \frac{\pi_\varphi(a_t|s_t)}{\pi_{\varphi_{\text{old}}}(a_t|s_t)}, \qquad (8)$$

where $\rho_t(\varphi)$ is the probability ratio and $\hat{A}_t$ is the estimated advantage estimator. $\hat{A}_t$ is computed by generalized advantage estimation (GAE) Schulman et al. (2015) taking the form of

$$\hat{A}_t = \delta_t + (\gamma\lambda)\delta_{t+1} + \cdots + (\gamma\lambda)^{T-t+1}\delta_{T-1}, \qquad (9)$$

where $\delta_t = V_t^{\text{targ}} - V(s_t) = r_t + \gamma V(s_{t+1}) - V(s_t)^3$. This objective can further be augmented by adding an entropy bonus to ensure sufficient exploration, as suggested in Schulman et al. (2017):

$$L(\varphi) = L^{\text{clip}}(\varphi) + c \cdot \text{entropy}[\pi_\varphi](s_t). \qquad (10)$$

Then $\varphi$ can be optimized by maximizing Eq. equation 10. Besides, a learned state-value function $V_\phi(\cdot)$ is leveraged to reduce variance in advantage estimators and can be trained by minimizing the following squared-error loss:

$$L(\phi) = \mathbb{E}_t[(V_\phi(s_t) - V_t^{\text{targ}})^2], \qquad (11)$$

where $V_t^{\text{targ}} = r_t + \gamma V(s_{t+1})$. Here, we use default hyperparameter settings suggested in Schulman et al. (2017): $\gamma = 0.99, \lambda = 0.95, \epsilon = 0.2, c = 0.01$.

# E    ADDITIONAL EXPERIMENTAL RESULTS

## E.1    EXPERIMENTS WITH A LONGER HORIZON

We conduct an experiment to train the target model for a longer time horizon. More concretely, RoBERTa-base is selected and the number of updates is double for each task as the original one shown in Table 8. We choose the polynomial learning rate scheduler as a representative example. As can be observed in Table 10, training longer can only slightly improve the performance. However, it still cannot match GNS, showing the efficiency and effectiveness of our proposed method.

## E.2    GENERALIZATION ON LANGUAGE TASKS

We also evaluate the generalization of GNS on language tasks, by transferring GNS trained on RoBERTa-base model to RoBERTa-large model. Results are reported in Table 11. We can see that GNS can still achieve relative satisfactory performance on large models and outperform the constant learning rate, even only trained on RoBERTa-base. This transferred performance also explains why only a few episodes are needed when fine-tuning the agent under RoBERTa-large models

---

[3]When we use $V(s_t)$ rather than $V_\phi(s_t)$, we refer to the value and it has no gradient w.r.t $\phi$.

Table 10: Comparison with the scheduler of a longer trajectory.

| Scheduler | CoLA | SST-2 | MRPC | QQP | STS-B | MNLI | QNLI | RTE | Avg. |
|---|---|---|---|---|---|---|---|---|---|
| Polynomial | 61.2 | 95.3 | 87.2 | 89.2 | 89.3 | 87.3 | 93.0 | 73.1 | 84.4 |
| Polynomial (longer) | 61.3 | **95.8** | 87.2 | 89.3 | 89.3 | 87.4 | 92.9 | 73.5 | 84.6 |
| GNS | **63.3** | **95.8** | **87.7** | **89.6** | **89.7** | **87.7** | **93.2** | **74.4** | **85.2** |

Table 11: Comparison between the constant learning rate and transferred GNS for RoBERTa-large.

| Scheduler | MRPC | STS-B | RTE | Avg. |
|---|---|---|---|---|
| Constant | $87.7 \pm 0.7$ | $90.1 \pm 0.3$ | $\mathbf{80.6 \pm 0.2}$ | 86.1 |
| GNS (transferred) | $\mathbf{87.9 \pm 0.8}$ | $\mathbf{91.1 \pm 0.2}$ | $79.5 \pm 0.2$ | **86.2** |

## E.3 SGD V.S. ADAM

As many papers also selected SGD as the standard optimizer for image classification tasks, we carry out an additional experiment to train a ResNet-34 on CIFAR10 with SGD. As shown in Table 12, we can observe that Adam offers comparable and slightly better performance than SGD, especially for the constant learning rate scheduling. It is consistent with conclusions in (Choi et al., 2019; Sivaprasad et al., 2020; Xiong et al., 2020) that Adam can serve as the most practical solution in training neural networks. Besides, GNS is still the best among all scheduling mechanisms with SGD, showing the effectiveness of our proposed method.

Table 12: Test accuracy with SGD and Adam as the base optimizer respectively on CIFAR10.

| Scheduler | SGD | Adam |
|---|---|---|
| Constant | $92.7 \pm 0.3$ | $93.0 \pm 0.4$ |
| Polynomial | $93.5 \pm 0.2$ | $93.5 \pm 0.3$ |
| Cosine | $93.4 \pm 0.4$ | $93.6 \pm 0.4$ |
| Hypergradient | $93.6 \pm 0.5$ | $93.7 \pm 0.6$ |
| SRLS | $93.6 \pm 0.4$ | $93.6 \pm 0.6$ |
| GNS | $\mathbf{94.1 \pm 0.3}$ | $\mathbf{94.3 \pm 0.5}$ |

## E.4 SENSITIVITY OF HYPERPARAMETERS

Table 13: Sensitivity of hyperparameters.

(a) Performace on different K with p fixed at 0.2.

| K | CIFAR10 | RTE |
|---|---|---|
| 1 | $92.5 \pm 0.6$ | $71.6 \pm 0.5$ |
| 5 | $93.0 \pm 0.4$ | $72.8 \pm 0.5$ |
| 10 | $\mathbf{94.3 \pm 0.5}$ | $\mathbf{74.4 \pm 0.3}$ |
| 20 | $93.7 \pm 0.4$ | $74.0 \pm 0.3$ |

(b) Performance on different p with K fixed at 10.

| p | CIFAR10 | RTE |
|---|---|---|
| 0.2 | $94.3 \pm 0.5$ | $74.4 \pm 0.3$ |
| 0.6 | $\mathbf{94.4 \pm 0.4}$ | $74.4 \pm 0.5$ |
| 1.0 | $\mathbf{94.4 \pm 0.4}$ | $\mathbf{74.6 \pm 0.4}$ |

Furthermore, we investigate the impact of two hyperparameters, the learning rate update interval $K$ and the validation probability $p$. Note that for CIFAR10, the real interval should multiply by 50, i.e., $K = 1$ denotes 50 training steps of CIFAR10. Two tasks on CIFAR10 and RTE are selected respectively from image classification and language understanding. It can be seen in Table 13a that when the learning rate is modified aggressively with $K = 1, 5$ or infrequently with $K = 20$, the scheduling would degrade the optimization performance, e.g., from 94.3 ($K = 10$) to 92.5 ($K = 1$). As to the validation probability, the performance of GNS is not very sensitive to different values of $p$

in Table 13b. Together with results of FRS in Table 4, we set $p$ to a relatively small value 0.2 in all experiments without much loss of performance while such a $p$ can reduce the time cost notably as discussed in Section 5.5.

## E.5    IMPACT OF MULTIPLE FACTORS

In this part, we investigate multiple factors such as the features, the encoding mechanism, and the hyperparameter. These factors affect the encoding and the most straightforward way to evaluate the encoding quality is to compare the performance of downstream tasks. In our paper the metric is just the performance of the target network trained with the neural scheduler. To investigate the encoding quality, we consider a list of schedulers with different encoding capability:

- SRLS

- SRLS + ALS (All Layer's States)

- SRLS + ALS + larger network, where the network is chosen from a 3-layer MLP with hidden size 32 (deeper), and a 2-layer MLP with hidden size 64 (wider).

- SRLS + ALS + larger network + tuned update interval K, K is chosen from $[1, 5, 10, 20]$.

- GNS

- GNS + GAT (Veličković et al., 2017)

Experiments are carried out on two datasets, CIFAR10 (ResNet34) and RTE (RoBERTa-base), and results are shown in Table 14. We can see that by using a super concatenated vector as the input, the performance is improved slightly, from 93.6 to 93.8 on CIFAR10 and 70.8 to 72.2 on RTE. A larger network also contributes to the performance gain to some degree but the improvement is limited, compared with the result of GNS, where the encoding scheme is significantly different from MLP networks. Note that performance with a tuned K remains the same and it indicates that our default update interval works the best. Moreover, after switching to a more powerful graph encoder, GAT, the performance can be further improved, in particular from 94.3 to 94.5 on CIFAR10.

To sum up, leveraging more information (SRLS -> SRLS + ALS) does enhance the scheduler. Using a better network architecture (a larger network) and hyperparameter configuration also helps but encoding with merely MLPs is not sufficient to make full use of all intermediate layers' information. By representing the target network as a directed graph and adopting an encoding scheme of GCN, the encoder quality improves and the corresponding scheduler is much better. With a stronger encoder GAT, the final performance can be further improved, which validates that the encoding quality matters for the learned scheduler.

Table 14: Comparison of schedulers with different configurations.

| Scheduler | CIFAR10 | RTE |
|---|---|---|
| SRLS | $93.6 \pm 0.6$ | $70.8 \pm 0.7$ |
| SRLS + ALS | $93.8 \pm 0.4$ | $72.2 \pm 0.5$ |
| SRLS + ALS + larger network | $93.9 \pm 0.4$ | $72.6 \pm 0.4$ |
| SRLS + ALS + larger network + tuned K | $93.9 \pm 0.4$ | $72.6 \pm 0.4$ |
| GNS | $94.3 \pm 0.5$ | $74.4 \pm 0.3$ |
| GNS + GAT | $\mathbf{94.5 \pm 0.4}$ | $\mathbf{74.7 \pm 0.3}$ |

## E.6    COMPARISON WITH THE ORIGINAL ROBERTA RESULTS

To make our results more convincing, we provide a comparison with the original results in Liu et al. (2019) as well as the standard practice (a warmup procedure followed by a linear decay) in Table 15. It can be observed that GNS performs consistently better on the dev set as well. we can tell that our proposed GNS still outperforms this standard method, indicating that the improvements are not due to the suboptimal learning rate schedules, but from our better learned policy.

Table 15: Development performance of RoBERTa-large on GLUE of different schedulers.

| Scheduler | CoLA | SST-2 | MRPC | QQP | STS-B | MNLI | QNLI | RTE | Avg. |
|---|---|---|---|---|---|---|---|---|---|
| *Large model:* | | | | | | | | | |
| Constant | 68.8 | 96.1 | 90.3 | 92.0 | 92.1 | 90.0 | 94.4 | 86.2 | 88.7 |
| Polynomial | 69.1 | 96.3 | 90.6 | 92.2 | 92.4 | 90.2 | 94.8 | 86.5 | 89.0 |
| Cosine | 69.0 | 96.4 | 90.5 | 92.2 | 92.3 | 90.1 | 94.6 | 86.7 | 89.0 |
| Hypergradient | 68.6 | 96.3 | 90.5 | 92.0 | 92.1 | 90.1 | 94.5 | 86.5 | 88.8 |
| SRLS | 67.8 | 95.9 | 89.7 | 91.6 | 92.1 | 89.9 | 94.2 | 86.0 | 88.4 |
| GNS | **70.5** | **97.1** | **91.2** | **92.7** | **92.9** | **90.5** | **94.9** | **87.4** | **89.6** |
| Liu et al. (2019) | 68.0 | 96.4 | 90.9 | 92.2 | 92.4 | 90.2 | 94.7 | 86.6 | 88.9 |
| Standard | 68.8 | 96.3 | 90.7 | 92.1 | 92.3 | 90.2 | 94.7 | 86.5 | 88.9 |

### E.7 COMPARISON WITH CMA-ES

In this part, we compare our GNS with a more effective hyperparameter tuning approach. More concretely, we leverage CMA-ES (Hansen, 2016) as another baseline, to search for the learning rate schedule. We choose a cosine decay method which leads to the best-performance results for almost all the tasks as the base scheduler when searching with CMA-ES. The corresponding search space is then expanded as shown in the following tables:

Table 16: Expanded search space for image classification

| Hyperparameter | Tuning range |
|---|---|
| $\alpha_{\max}$ | LogUniform$[10^{-3}, 10^{-1}]$ |
| $\mu$ | LogUniform$[10^{-3}, 10^{-1}]$ |
| $\eta_2$ | Uniform$[0.5, 1.0]$ |

Table 17: Expanded search space for language understanding

| Hyperparameter | Tuning range |
|---|---|
| $\alpha_{\max}$ | LogUniforrm$[10^{-5}, 10^{-4}]$ |
| $\eta_1$ | Uniform$[0.0, 0.1]$ |

50 evaluations are conducted in total for CMA-ES and we also train GNS for 50 episodes for a fair comparison. Results are demonstrated in Figure 5. We can see that CMA-ES can find a satisfactory hyperparameter configuration quickly in the beginning, but our GNS can reach a better final performance. When we consider the whole search trajectory, it is still safe to say that the proposed GNS is a relatively efficient and effective method even compared with the more powerful baseline CMA-ES.

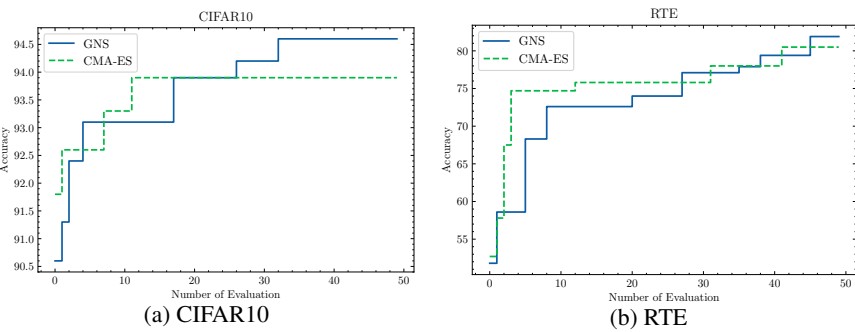

(a) CIFAR10 (b) RTE

Figure 5: Search performance of GNS and CMA-ES.

### E.8 ABLATION STUDY ON THE CHOICE OF RAW FEATURES

In order to characterize the training dynamics, we picked some common features adopted in (Xu et al., 2017; 2019), including the mean and variance of weights, biases, and the mean and variance of absolute gradient values, as well as global dynamics of the previous learning rate and the average training loss. We can roughly group them into three categories: parameter state, gradient state, and global state. To investigate the influence of different features, we mask one of these three types of features and only use the remaining two as the raw input each time. Then we use different input features to train GNS on CIFAR10 and RTE. Detailed results are presented in the following table. It can be observed that all three categories of features benefit the quality of GNS. Parameter and gradient information is more important than the global state since the global state can be reflected by the reward and the action. This ablation study validates our choice of these features. In the future, it is also interesting to integrate more useful features to further improve the performance, such as the accumulated momentums which are used in adaptive optimizers like Adam.

Table 18: Performance of GNS with different input features.

| Scheduler | CIFAR10 | RTE |
|---|---|---|
| GNS w/o parameter state | $93.8 \pm 0.6$ | $73.3 \pm 0.5$ |
| GNS w/o gradient state | $93.7 \pm 0.4$ | $73.5 \pm 0.4$ |
| GNS w/o global state | $94.1 \pm 0.6$ | $74.0 \pm 0.3$ |
| GNS | $\mathbf{94.3 \pm 0.5}$ | $\mathbf{74.4 \pm 0.3}$ |

### E.9 COMPARISON OF OTHER FORMS OF ACTION

We have tried two other forms of action, adding a residual and directly predicting the learning rate. For adding a residual, since the learning rate for different tasks might have different magnitudes, we have to design some additional hyperparameters elaborately for each task to constrain the output residual to a specific range. As to predicting the learning rate, it is very unstable and usually breaks the optimization. We also implemented the action of adding a residual by constraining the output to the range $[-10^{-5}, 10^{-5}]$, and the action of predicting the learning rate by replacing the activation function with Sigmoid on CIFAR10, and the corresponding results are shown as follows:

Table 19: Performance of GNS with different forms of action.

| Scheduler | CIFAR10 |
|---|---|
| GNS (adding a residual) | $90.7 \pm 0.6$ |
| GNS (predicting the learning rate) | $10.0 \pm 0.0$ |
| GNS (multiplying by a factor) | $\mathbf{94.3 \pm 0.5}$ |

We can see that adding a residual is insufficient to beat our original design of multiplying by a factor. For predicting the learning rate, it is hard to obtain a useful scheduler and the training of CIFAR10 failed and was stuck with only 10% accuracy. However, it is possible that by fine-tuning the hyperparameters for these two forms of action, the obtained scheduler will work. But multiplying by a factor is the most stable action and needs little tuning effort and that's why we choose this action.

### E.10 EVALUATION ON MLPS

We include an evaluation of training a 3 layer MLP on Fashion MNIST with the hidden size of 128 and we report the performance of different optimizers in Table 20. It can be seen that GNS is also applicable to such a simple scenario, with the best test accuracy of 89.8.

Table 20: Test accuracy of a 3-layer MLP on Fashion MNIST.

| Scheduler | Fashion MNIST |
|---|---|
| Constant | $88.8 \pm 0.2$ |
| Polynomial | $89.3 \pm 0.3$ |
| Cosine | $89.3 \pm 0.2$ |
| Hypergradient | $89.0 \pm 0.2$ |
| SRLS | $89.5 \pm 0.4$ |
| GNS | $\mathbf{89.8 \pm 0.3}$ |

## E.11 TRAINING LOSS CURVE

We have included the training loss curve for Fashion MNIST and CIFAR10 in Figure 6. Since for the GLUE benchmark tasks, final performance is usually the focus in the literature and the loss and the evaluation metric do not necessarily correlate, we do not show their loss curves.

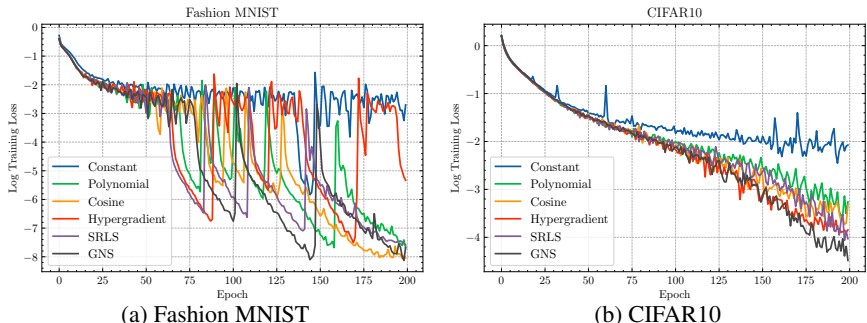

(a) Fashion MNIST      (b) CIFAR10

Figure 6: Training loss curves for Fashion MNIST and CIFAR10.

