# OpenReview forum: "Learning to Schedule Learning rate with Graph Neural Networks"
_ICLR.cc/2022/Conference — ICLR 2022 Poster_

### Official Review · Reviewer_NP67 · 2021-10-30

**Correctness:** 3
**Technical Novelty And Significance:** 3
**Empirical Novelty And Significance:** 3
**Recommendation:** 6
**Confidence:** 3

**Main Review:**

Pros: The proposed approach for the learning rate scheduling is novel. It can achieve better performance than baselines.

Issues:
1. Can the proposed method be applicable to simpler models, such as multilayer perceptrons (MLPs)?

2. For a graph constructed based on a network model, what are the features of nodes?  How to keep the dimensions of node features consistent?

3. The training loss curve should be displayed for better comparison.

4. In the experiment, only some simple hand-designed scheduling mechanisms are compared. The author should compare some baseline methods based on reinforcement learning, such as [1,2].

[1] Zhen Xu, Andrew M Dai, Jonas Kemp, and Luke Metz. Learning an adaptive learning rate schedule.
arXiv preprint arXiv:1909.09712, 2019.

[2] Chang Xu, Tao Qin, Gang Wang, and Tie-Yan Liu. Reinforcement learning for learning rate control.
arXiv preprint arXiv:1705.11159, 2017.

**Summary Of The Paper:**

In this paper, the authors study the learning rate scheduling for optimization in training deep neural networks.
It proposes graph-network-based scheduler (GNS) to learn a scheduling mechanism for learning rate.
GNS encodes the neural network by GNN and trains an agent to control the learning rate.
Extensive experiments are conducted to demonstrate the effectiveness of the proposed method.

**Summary Of The Review:**

In general, the paper is written well and easy to follow. However, there exist aforementioned concerns that the authors need to address.

---

> ### Author Response · Authors · 2021-11-16
> **Response to Reviewer NP67**
>
> We appreciate that you have made such valuable comments and suggestions to improve our work. We present our detailed response to your concerns below.
>
> **[Q1: Applying GNS to simpler models]**
> Thanks for the suggestion of evaluation on simpler models.  We include an evaluation of training a 3 layer MLP on Fashion MNIST with the hidden size of 128 and we report the performance of different optimizers in Table 20 in Appendix E.10. It can be seen that GNS is also applicable to such a simple scenario, with the best test accuracy of 89.8.
>
> |  Scheduler | Fashion MNIST |
> | :---       |    :----:           |
> | Constant      | 88.8  ± 0.2     |
> | Polynomial   | 89.3  ± 0.3     |
> | Cosine   | 89.3 ± 0.2    |
> | Hypergraident  | 89.0 ± 0.2 |
> | SRLS   | 89.5 ± 0.4       |
> | GNS   | **89.8 ± 0.3**   |
>
> **[Q2: Features of nodes and how to keep the dimension consistent]**
> In order to characterize the training dynamics, we picked some common summarized features adopted in [1,2], including the mean and variance of weights, biases, and the mean and variance of absolute gradient values, as well as global dynamics of the previous learning rate and the average training loss. We leverage the same choice of raw features for all different network architectures, and thus the dimension of features for each node is consistent across all the network structures.
>
> We also conducted an ablation study to account for the choice of these values as the raw features. In detail, we can roughly group them into three categories: parameter state, gradient state, and global state. To investigate the influence of different features, we mask one of these three types of features and only use the remaining two as the raw input each time. Then we use different input features to train GNS on CIFAR10 and RTE. Detailed results are presented in Table 18 in Appendix E.8. It can be observed that all three categories of features benefit the quality of GNS. Parameter and gradient information is more important than the global state since the global state can be reflected by the reward and the action. This ablation study validates our choice of these features. In the future, it is also interesting to integrate more useful features to further improve the performance, such as the accumulated momentums which are used in adaptive optimizers like Adam.
>
> |  Scheduler     | CIFAR10 | RTE     |
> | :---        |    :----:   |          :----: |
> | GNS w/o parameter state      | 93.8 ± 0.6       | 73.5  ± 0.5 |
> | GNS w/o gradient state  | 93.7  ± 0.4       | 73.5  ± 0.4    |
> | GNS w/o global state  | 94.1  ± 0.6     | 74.0  ± 0.3   |
> | GNS   | 94.3  ± 0.5       | 74.4  ± 0.3     |
>
> **[Q3: The training loss curve]**
> We have included the training loss curve for Fashion MNIST and CIFAR10 in Figure 6 in Appendix E. Since for the GLUE benchmark tasks, final performance is usually the focus in the literature and the loss and the evaluation metric do not necessarily correlate, we do not show their loss curves.
>
> **[Q4: Comparison of RL-based methods]**
> In our paper, we compared our proposed method with a RL-based method, SRLS, which is exactly the method in [1]. For the method in [2], as the source code is not provided, we re-implemented it by ourselves and found that it could not lead to a stable scheduler (failing to train a ResNet on CIFAR10 well, with only a random guessing accuracy 10%) as the action is predicting the learning rate directly. Thus, we only reported the performance of the better RL-based scheduler, SRLS in all experiments. We have demonstrated advantages of GNS such as efficiency and effectiveness over SRLS in our paper.
>
> [1] Xu, Zhen, et al. "Learning an adaptive learning rate schedule." arXiv preprint arXiv:1909.09712 (2019).
>
> [2] Xu, Chang, et al. "Reinforcement learning for learning rate control." arXiv preprint arXiv:1705.11159 (2017).

---

### Official Review · Reviewer_woPs · 2021-10-30

**Correctness:** 4
**Technical Novelty And Significance:** 3
**Empirical Novelty And Significance:** 3
**Recommendation:** 8
**Confidence:** 4

**Main Review:**

Strengths:

- The authors has done a great job by employing the techniques from RL and graph neural networks to formulate learning rate schedule and find the optimal trajectory. Finding the optimal schedule of learning rate is an interesting and important practical problem which can help improve the performance of the models in all machine learning community.

- Using graph neural networks for computing the features of the nodes is an innovative idea, because it offers two benefits that the other methods lack: 1- it can use all the information in the middle layers as opposed to the last layer, 2- the learned graph convolutional network (GCN) can be reused and generalized to new networks, so this makes it architecture-agnostic.

- The experiments are convincing and complete. It is clear to see the benefits of GNS over all other methods. The authors has shown the performance improvement in different tasks for image classification and language understanding, analyzed the generalization of GNS between different networks, compared the run times, and done an ablation study to show the importance of different parts of the model.

- The paper is well-written and easy to follow and understand.

Weaknesses:

- The authors has used GCN as the model of choice for GNNs. But there is no explanation on why they choose this. I wonder if the authors have also considered other models such as graph attention networks (GAT)? The attention mechanisms have proven effective in performance improvement in many tasks. It would be great if the authors check to see if other GNNs can even outperform GCNs in this specific problem.




**Summary Of The Paper:**

This paper formulates the learning the schedule of learning rate as a reinforcement learning (RL) problem and finds the optimal schedule which outperforms the baseline scheduling algorithms such as constant learning rate, polynomial or cosine decay, warmup and restart, gradient-based method like Hypergradient, and another learning-based method SRLS. The paper proposes Graph-Network-based Scheduler (GNS). Rather than looking at the last layer or concatenating different layers of the network, GNS uses graph convolutional networks (GCN) to do message passings over the computational graphs of the model, and as a results, are more efficient, informative, and architecture-agnostic which can easily generalize to other network architectures. The authors has shown that GNS outperforms all other learning rate scheduling algorithms and achieves superior performance in various tasks.

**Summary Of The Review:**

I believe the problem addressed in this paper (how to learn optimal learning rate schedule) is very important to the machine learning community, and the techniques used to solve it (RL and GNNs) are powerful and effective. The authors has formulated the problem as an RL problem, which is the most correct way to do it, and has also thought how to make it architecture-agnostic, generalizable, and versatile in using the hidden layers' information by message passing through the computational graphs. Overall, it seems that this formulation and solution is the best one can do to learn the optimal learning rate schedule, which is validated by the experimental results and the ablation study. Therefore, I believe this paper is very interesting and I recommend it for acceptance.

---

> ### Author Response · Authors · 2021-11-16
> **Response to Reviewer woPs**
>
> We are happy to know that you recognize the significance of our work and acknowledge our method is novel and effective.
> We provide our reply to your question below.
>
> **[Q1: Incorporate GAT into the scheduler]**
> We use GCN in the paper due to its effectiveness and simplicity compared with other GNNs. For your question about using a more powerful graph network like GAT [1], it is a good suggestion and we carried out an experiment to replace the current GCN model with the GAT model on CIFAR10 and RTE. Detailed performance is shown in Table 14 in Appendix E.5. We can observe from Table 14 that GAT does improve the encoding quality and lead to a more effective GNS, in particular, from 94.3 to 94.5 on CIFAR10, and from 74.4 to 74.7 on RTE. The improvement is not very obvious and we assume that the current raw features constrain GAT and do not fully leverage the encoding capability of GAT. In the future, we will further investigate the choice of raw features as well as underlying encoding models.
>
> |  Scheduler     | CIFAR10 | RTE     |
> | :---       |    :----:           |  :----:           |
> | GNS   | 94.3  ± 0.5       | 74.4  ± 0.3     |
> | GNS + GAT   | **94.5 ± 0.4**       | **74.7 ± 0.3**       |
>
> [1] Veličković, Petar, et al. "Graph attention networks." arXiv preprint arXiv:1710.10903 (2017).

---

> > ### Comment · Reviewer_woPs · 2021-11-29
> > **Concern addressed**
> >
> > Thanks for your response and addressing my comment. It is interesting to see GAT helps improve the accuracy. I have no other concerns and would like to keep my score (8).

---

> > > ### Author Response · Authors · 2021-11-30
> > > **Thanks for your reply**
> > >
> > > Dear Reviewer woPs,
> > >
> > > Thanks again for your acknowledge of our work as well as your helpful suggestion. We will explore the impact of different encoding networks extensively in the future.
> > >
> > > Best,
> > >
> > > Paper124 Authors

---

### Official Review · Reviewer_RAQV · 2021-11-02

**Correctness:** 3
**Technical Novelty And Significance:** 2
**Empirical Novelty And Significance:** 3
**Recommendation:** 6
**Confidence:** 4

**Main Review:**

1. Finding a good learning rate scheduler is an interesting topic. This paper proposes a new way to search for a good learning rate by exploiting the neural network structure with a graph.
2. In practice, the mean and variance of the weights, biases, and absolute gradient values, etc. are used as the node features. Why choose them as the features?
3. What is the learning rate for the training of GNS? How to set this learning rate? How this learning rate will affect the performance? Those questions should be investigated.
4. Why not use some other forms of action, for example, adding a residual?
5. I recommend reporting the performance on ImageNet to make it more convincing.
6. The full form of some abbreviations is not declared, which makes it hard to understand for some readers, for example, MRPC, RTE, and STS-B.

**Summary Of The Paper:**

The paper proposes a Graph-based learning rate scheduler for neural network optimization. The graph is constructed based on the network structures and trained with reinforcement learning. The method is evaluated based on Image Classification and Natural Language Understanding and achieves superior performance than baselines.

**Summary Of The Review:**

In summary, the paper proposes a novel and effective way to learn a learning rate scheduler. However, some of the investigations such as the node feature and the meta-learning rate are missing.

-- post-rebuttal

My concerns are well-addressed in the rebuttal. I will keep my score.

---

> ### Author Response · Authors · 2021-11-16
> **Response to Reviewer RAQV (1/2)**
>
> We are grateful for your careful and valuable comments, and our detailed response to your questions is presented as follows.
>
> **[Q1: Interesting topic]**
> We appreciate that you acknowledge our effort in this work and we have made some modifications to refine the paper based on all the reviewers’ suggestions.
>
> **[Q2: The choice of raw features]**
> We are sorry that we did not pay much attention to the choice of raw features. In order to characterize the training dynamics, we picked some common features adopted in [1,2], including the mean and variance of weights, biases, and the mean and  variance of absolute gradient values, as well as global dynamics of the previous learning rate and the average training loss. We can roughly group them into three categories: parameter state, gradient state, and global state. To investigate the influence of different features, we mask one of these three types of features and only use the remaining two as the raw input each time. Then we use different input features to train GNS on CIFAR10 and RTE. Detailed results are presented in Table 18 in Appendix E.8. It can be observed that all three categories of features benefit the quality of GNS. Parameter and gradient information is more important than the global state since the global state can be reflected by the reward and the action. This ablation study validates our choice of these features. In the future, it is also interesting to integrate more useful features to further improve the performance, such as the accumulated momentums which are used in adaptive optimizers like Adam.
>
> |  Scheduler     | CIFAR10 | RTE     |
> | :---        |    :----:   |          :----: |
> | GNS w/o parameter state      | 93.8 ± 0.6       | 73.5  ± 0.5 |
> | GNS w/o gradient state  | 93.7  ± 0.4       | 73.5  ± 0.4    |
> | GNS w/o global state  | 94.1  ± 0.6     | 74.0  ± 0.3   |
> | GNS   | 94.3  ± 0.5       | 74.4  ± 0.3     |
>
> **[Q3: The learning rate for training GNS]**
> For the learning rate in the PPO algorithm, we adopt the grid search to tune it within [0.001, 0.005] for both the actor and the critic, which is suggested in [1]. We found that the recommended setting with $\alpha_{\text{actor}}=0.001$ and $\alpha_{\text{critic}}=0.005$ worked well to train GNS under different downstream tasks. Therefore, we just use this learning rate configuration ( $\alpha_{\text{actor}}=0.001$ and $\alpha_{\text{critic}}=0.005$) in all our experiments including computer vision and NLP tasks. Fine-tuning the learning rate with a larger space and adopting a more efficient search algorithm are likely to further improve the performance but they are relatively expensive so we leveraged the introduced simple yet effective grid search method within the suggested search range.
>
> **[Q4: Other forms of action]**
> We have tried two other forms of action, adding a residual and directly predicting the learning rate. For adding a residual, since the learning rate for different tasks might have different magnitudes, we have to design some additional hyperparameters elaborately for each task to constrain the output residual to a specific range. As to predicting the learning rate, it is very unstable and usually breaks the optimization. We also implemented the action of  adding a residual by constraining the output to the range  $[-10{^-5}, 10{^-5}]$, and the action  of predicting the learning rate by replacing the activation function with sigmoid on CIFAR10,  and the corresponding results are shown as follows:
>
> |  Scheduler | CIFAR10 |
> | :---       |    :----:           |
> | GNS (adding a residual)     | 90.7 ± 0.6       |
> | GNS (predicting the learning rate)| 10.0 ± 0.0      |
> | GNS (multiplying by a factor)|  **94.3 ± 0.5**  |
>
> We can see that adding a residual is insufficient to beat our original design with multiplying by a factor. For predicting the learning rate, it is hard to obtain a useful scheduler  and the training of CIFAR10 failed and was stuck with only 10% accuracy.  However, it is possible that by fine-tuning the hyperparameters for these two forms of action, the obtained scheduler will work. But multiplying by a  factor is the most stable action and needs little tuning effort and  that’s why we choose this action definition.
>
> (continued)

---

> > ### Author Response · Authors · 2021-11-16
> > **Response to Reviewer RAQV (2/2)**
> >
> > **[Q5: ImageNet results]**
> > To make our method more convincing, we have added the performance of different schedulers on ImageNet in Table 3 of transferred performance of GNS. Still, we adopt the best hyperparameter configuration from ResNet34 on CIFAR10 to train the same model structure on ImageNet. For GNS, we just apply the policy network trained under CIFAR10 to adjust learning rate for training on ImageNet. It can be seen that our GNS works consistently well and leads to the best test accuracy. In particular, transferred performance of GNS is 73.1% and is better than the best baseline of 71.6%.
> >
> > |  Scheduler | CIFAR10 -> ImageNet |
> > | :---       |    :----:           |
> > | Constant      | 68.9       |
> > | Polynomial   | 70.3       |
> > | Cosine   | 70.5     |
> > | Hypergraident  | 70.0 |
> > | SRLS   | 71.6       |
> > | GNS   | **73.1**   |
> >
> > **[Q6: Abbreviations]**
> > We are sorry that these abbreviations caused confusion for you. They are the abbreviated name for each dataset in the GLUE benchmark tasks, and it is a convention to directly use the abbreviations to represent a specific task. Here we list the full name for your reference:
> > - CoLA: The Corpus of Linguistic Acceptability
> > - SST-2: The Stanford Sentiment Treebank
> > - MRPC: Microsoft Research Paraphrase Corpus
> > - QQP: Quora Question Pairs
> > - STS-B: Semantic Textual Similarity Benchmark
> > - MNLI: The Multi-Genre Natural Language Inference
> > - QNLI: The Question Natural Language Inference
> > - RTE: Recognizing Textual Entailment
> >
> > You can also check [this website](https://gluebenchmark.com/tasks) for more details.
> >
> > [1] Xu, Zhen, et al. "Learning an adaptive learning rate schedule." arXiv preprint arXiv:1909.09712 (2019).
> >
> > [2] Xu, Chang, et al. "Reinforcement learning for learning rate control." arXiv preprint arXiv:1705.11159 (2017).

---

> > > ### Comment · Reviewer_RAQV · 2021-11-28
> > > **Additional questions**
> > >
> > > Thanks for the clarification and additional experiments.
> > >
> > > Q5: Why the policy network trained under CIFAR10 is used in the experiment? Is it fair to compare with other methods in this way since additional data is used for GNS?

---

> > > > ### Author Response · Authors · 2021-11-28
> > > > **Reply to the new question**
> > > >
> > > > Thank you for your reply. We chose to leverage the policy network trained under CIFAR10 to evaluate the generalization of GNS to different network structures and different datasets, which is a similar evaluation metric adopted in [1]. It is still a fair comparison when we used the policy network trained under CIFAR10, as hyperparameters of traditional methods (constant, polynomial, cosine, hypergradient) were also tuned on CIFAR10, and the RL-based method SRLS used the policy network trained under CIFAR10 as well. In other words, all the schedulers were tuned for the best performance on CIFAR10, and then leveraged directly to tasks with different networks (e.g., VGG) or different datasets (e.g., CIFAR100 and ImageNet). Thus, CIFAR10 was used for all methods and should not be regarded as additional data. The performance gain of GNS is due to one of advantages of our proposed method, which is great generalization to different scenarios.
> > > >
> > > > [1] Xu, Zhen, et al. "Learning an adaptive learning rate schedule." arXiv preprint arXiv:1909.09712 (2019).

---

> > > > ### Author Response · Authors · 2021-11-29
> > > > **Reply to Reviewer RAQV**
> > > >
> > > > Dear Reviewer RAQV,
> > > >
> > > >
> > > > We hope that our reply has solved your concern. Please let us know if you have further questions and we are willing to answer them and provide more information.
> > > >
> > > > Best,
> > > >
> > > > Paper124 Authors

---

### Official Review · Reviewer_RKqo · 2021-11-02

**Correctness:** 3
**Technical Novelty And Significance:** 3
**Empirical Novelty And Significance:** 2
**Recommendation:** 6
**Confidence:** 3

**Main Review:**

# Strengths:

* The improvements in the paper are intuitive and consistent across the benchmarks considered. The paper makes an effective presentation of the idea and the experiments are largely sound.

* It is great that the paper improves realistic large models in settings such as RoBERTa fine-tuning.

* It is natural and useful, in my opinion, that the authors utilize the graph structure of the computational graph.

# Weaknesses:

*  The training episodes for the PPO algorithm provide a serious bottleneck for searching for a learning rate scheduling policy. The authors address the problem in Section 5.5, however they do not say how much time it took for the CIFAR experiments. Could you clarify? For example, I am missing the details of what are $T$ and $K$ in Algorithm 1 for the CIFAR experiments. In Section 6 the authors also acknowledge that limitation, but they do not discuss how the potential future direction might look like. It is not clear to me how this method could be useful for long time horizons.

* Focusing on the current experiments, it seems that the empirical argument the paper makes is that using GNS is more optimal than doing a grid search. What if we expand the space and use a more efficient search methods in the hyperparameter space? For example, a natural baseline seems to be CMA-ES (https://arxiv.org/abs/1604.00772). It would be interesting if there is something to say/ show about comparing your method to more efficient alternatives of grid search.

* Is there an ablation on the choices of the raw features $x_v$ used in the GCN? It would be useful to take a look at such a study for future work with GNS.

* I am not sure I understand the last sentence of Section 4, "During inference, we first sample...". Could you please clarify what the procedure is and how it is used in your experiments, because I don't think I see that described in Algorithm 1?


## Minor:

* In Equation 1 the matrix $\mathbf{W}^l$ is not formally defined.

* At the end of page 4, please use different enumeration for the steps in the hierarchical GNNs, because (a,b,c) is used in Figure 1 and it becomes difficult to read the text.

* In the top of page 5, "directly or with slight fine-tuning": this is vague, could you be more precise what the fine-tuning is?

* In the beginning of Section 4.2: the rest components <- the remaining components.

* Page 6: estimated advantage estimator <- estimated advantage (redundant "estimator").

* Algorithm 1 on page 13: Why is Eq. (1) an optimization step, i.e. are you sure you are referencing the right equation?; Section 3.3 is not present in the main text (perhaps a typo?).

* Page 16: e can see that GNS <- We can see that GNS (typo).

**Summary Of The Paper:**

This paper proposes a Graph Network-based Scheduler (GNS) method that learns a learning rate scheduling policy for problems with moderate time horizons for optimization. The novelty of the paper consist of representing the optimized neural network as a graph and using a policy that takes GCN representations of the architecture and its weights in a PPO algorithm. The paper also introduces strategies for validation that allow for more efficient searching of the policy. There are extensive experiments on training ResNets on CIFAR-10 and fine-tuning RoBERTA on GLUE. Additional benefits are demonstrated with transferring the learned policies to novel architectures or datasets. Ablation studies underscore the components in the GNS method.

**Summary Of The Review:**

My initial recommendation is 5: marginally below the acceptance threshold. I believe that this paper has great potential to become useful to the community. However, in my opinion there are some important questions that need to be clarified. I have listed my comments/ thoughts and suggestions in the Main Review. If you could please address my comments, I could consider increasing my score.

-- post-rebuttal

Thank you for your response. I will raise my score to 6.

---

> ### Author Response · Authors · 2021-11-16
> **Response to Reviewer RKqo (1/2)**
>
> We are happy that you think our work has great potential and we appreciate your insightful questions and our detailed response is presented below.
>
> **[Q1: Training details of CIFAR10]** It took 1.9 hours on average to finish one episode of CIFAR10 on a single NVIDIA 1080Ti GPU. And the details of T and K for image classification tasks are presented below. If we conduct the search sequentially, then time is 1.9*30=57 hours. Instead, we implemented the algorithm in parallel with 8 GPUs, which decreased the cost to only 8 hours.
>
> |   | Fashion MNIST |  CIFAR10 |
> | :---       |    :----:           |  :----:           |
> | # updates T      | 78200       | 62600 |
> | Decision interval K   | 500       |  500|
>
> Beside, to further reduce the time cost and to enable GNS in long-horizon tasks, a possible solution is to early stop the episode when the performance degrades obviously, using a similar idea of successive halving in Hyperband [1]. We have added it to Section 6 as a future direction.
>
> For your question about extension to tasks with longer horizons, the current mechanism can be applied to tasks with any time horizon. In detail, when training the scheduler with a reinforcement learning method, we set the episode length to the total training steps of the target problem. For example, if we want to train a network on CIFAR10 for 300 epochs instead of 200 epochs in the paper, we set the episode length to 300 epochs. On the other hand, if you mean applying the scheduler trained with a shorter horizon to the task with a longer one, it is related to another aspect of generalization and  we did not investigate the generalization to the longer horizon in the current version. However, it is also an interesting future direction that can be explored.
>
> **[Q2: Comparison with more effective search methods]** Thank you for your suggestion of a comparison with some more efficient search algorithms. Thus, we leverage CMA-ES [2] as another baseline to search for the learning rate schedule. We choose a cosine decay method which leads to the best-performance results for almost all the tasks as the base scheduler when searching with CMA-ES. The corresponding search space for image classification and language tasks is then expanded as shown in the following table:
>
> |Hyperparameter | Tuning range |
> | :---       |  :---            |
> |$\alpha_\text{max}$ | LogUniform $[10^{-3}, 10^{-1}]$|
> |$\mu$ | LogUniform $[10^{-3}, 10^{-1}]$ |
> |$\eta_2$ | Uniform $[0.5,1.0]$|
>
> |Hyperparameter | Tuning range |
> | :---       |    :---          |
> |$\alpha_\text{max}$ | LogUniform $[10^{-5}, 10^{-4}]$|
> |$\eta_1$ | Uniform $[0.0,0.1]$|
>
> 50 evaluations are conducted in total for CMA-ES and we also train GNS for 50 episodes for a fair comparison. Results are demonstrated in Figure 5 in Appendix E.7. We can see that CMA-ES can find a satisfactory hyperparameter configuration very quickly in the beginning, but our GNS can reach a better final performance. When we consider the whole search trajectory, it is still safe to say that the proposed GNS is a relatively efficient and effective method even compared with the more powerful baseline CMA-ES.
>
> **[Q3: Ablation study on the choice of raw features]**
> We are sorry that we did not pay much attention to the choice of raw features. In order to characterize the training dynamics, we picked some common features adopted in [3, 4], including the mean and variance of weights, biases, and the mean and  variance of absolute gradient values, as well as global dynamics of the previous learning rate and the average training loss.
>
> We can roughly group them into three categories: parameter state, gradient state, and global state. To investigate the influence of different features, we mask one of these three types of features and only use the remaining two as the raw input each time. Then we use different input features to train GNS on CIFAR10 and RTE. Detailed results are presented in Table 18 in Appendix E.8. It can be observed that all three categories of features benefit the quality of GNS. Parameter and gradient information is more important than the global state since the global state can be reflected by the reward and the action. This ablation study validates our choice of these features. In the future, it is also interesting to integrate more useful features to further improve the performance, such as the accumulated momentums which are used in adaptive optimizers like Adam.
>
> |  Scheduler     | CIFAR10 | RTE     |
> | :---        |    :----:   |          :----: |
> | GNS w/o parameter state      | 93.8 ± 0.6       | 73.5  ± 0.5 |
> | GNS w/o gradient state  | 93.7  ± 0.4       | 73.5  ± 0.4    |
> | GNS w/o global state  | 94.1  ± 0.6     | 74.0  ± 0.3   |
> | GNS   | 94.3  ± 0.5       | 74.4  ± 0.3     |
>
> (continued)

---

> > ### Author Response · Authors · 2021-11-16
> > **Response to Reviewer RKqo (2/2)**
> >
> > **[Q4: Inference of the learned optimizer]**
> > We apologize for the confusion. This sentence describes how we apply the learned GNS to downstream tasks, so it is not included in the training procedure  in  Algorithm 1. During the inference, i.e., when using GNS to adjust learning rate for target problems, we first need to choose an initial learning rate. As  GNS is trained under a specific distribution of the learning rate, we just sample some values based on this distribution. For example, on CIFAR10, we only use three learning rates from $[10^{-3}, 10^{-2}, 10^{-1}]$. And we run the training of ResNet on CIFAR10 using these three initial learning rates with GNS independently for $T_\text{pre}=1000$ steps, and compute corresponding state values for each initial learning rate. Finally, we select the learning rate with the largest state value to complete the training, which indicates the current chosen state is the most promising according to the value network. We hope this explanation can solve your concern and we understand that the inference might not be a proper word and we have modified the sentence to make it more clear.
> >
> > **[Q5: Minor questions]**
> > Thank you for your careful reading. We reply to each point below:
> > - $\mathbf{W}^l$ is the parameters  of the $l$-th layer of  GCN, and we add this description to the paper.
> > - Thanks for your suggestion and we have modified the enumeration.
> > - Here fine-tuning means that for a new task, we can initialize the parameters of the critic and the actor as those of models that have already been trained from a previous task. For example, if the target problem is CIFAR100, and we have trained the critic and the actor under the task of CIFAR10, then we can initialize the parameters as $\phi_\text{cifar10}, \varphi_\text{cifar10}$, and train them on CIFAR100 for a few episodes.
> > - We have corrected this grammar error.
> > - We have deleted the redundant word.
> > - We have corrected the typo in Algorithm 1 and on page 16.
> >
> > [1] Li, Lisha, et al. "Hyperband: A novel bandit-based approach to hyperparameter optimization." The Journal of Machine Learning Research 18.1 (2017): 6765-6816.
> >
> > [2] Hansen, Nikolaus. "The CMA evolution strategy: A tutorial." arXiv preprint arXiv:1604.00772 (2016).
> >
> > [3] Xu, Zhen, et al. "Learning an adaptive learning rate schedule." arXiv preprint arXiv:1909.09712 (2019).
> >
> > [4] Xu, Chang, et al. "Reinforcement learning for learning rate control." arXiv preprint arXiv:1705.11159 (2017).

---

> ### Author Response · Authors · 2021-11-28
> **Thanks for your reply**
>
> Dear Reviewer RKqo,
>
> We are happy to know that our response solved your concern and thank you again for your constructive and valuable comments.
>
> Best,
>
> Paper124 Authors

---

### Official Review · Reviewer_TPKw · 2021-11-03

**Correctness:** 3
**Technical Novelty And Significance:** 2
**Empirical Novelty And Significance:** 2
**Recommendation:** 6
**Confidence:** 4

**Main Review:**

Strengths:
-- The paper explores an interesting idea -- using GCN to encode all model states to better learn a learning rate schedule for a given model.

Weaknesses:
-- The novelty of the paper is limited.
-- The paper fails to provide convincing evidence that the suboptimal accuracy of existing methods is truly a result of using the final layer state, e.g., is it possible that the RL-based optimization scheme itself is suboptimal?
-- The evaluation of image classification tasks is conducted on toy datasets such Fashion-MNIST/CIFAR10. To be more convincing, the paper should include an evaluation of larger datasets such as ImageNet.
-- Evaluation on RoBERTa-base/large on GLUE datasets appear to be using sub-optimal baselines.
-- Might not be easy to apply in practice.


**Summary Of The Paper:**

This paper introduces a data-driven approach to automatically learn the learning rate schedule for a given dataset/task. The paper claims that existing RL-based adaptive learning rate schedule methods are insufficient to work well for large models such as RoBERTa because they only use model states from the last layer. The paper then proposes to use all layers' hidden states as feature vectors and uses graph neural network GCN to encode those features. The paper claims that such a feature encoding scheme allows reinforcement learning (e.g., PPO) to better learn a learning rate schedule, especially for large models. The paper evaluates its proposed method on both image classification (e.g., Fashion-MNIST and CIFAR-10) and language understanding tasks (finetuning RoBERTa on GLUE) and shows that the proposed method is able to achieve better final accuracy than some heuristic-based learning rate schedules.

**Summary Of The Review:**

The paper explores an interesting idea of using graph neural networks to better encode model states, with a hope to improve the existing RL-based meta-learning method. Despite showing some promising results, there are quite a few concerns about this work.

First, the RL-based scheme (e.g., PPO) is very similar to Xu et al. 2019. Therefore, the remaining novelty is really on changing FFN to GCN to encode more model hidden states.

Second, it seems there could be multiple factors that impact the final accuracy, such as the encoding scheme (e.g., GCN vs. other encoders), the features (last layers vs. all intermediate states), the optimization mechanism (RL vs. alternative methods), the choice of hyperparameters. However, the paper jumps to the conclusion that the encoding scheme and features are suboptimal without giving enough evidence to show that they are indeed the culprit. The paper also changes multiple factors at once, including tuning additional hyperparameters, making it difficult to tell where the improvements actually come from. The end-to-end results are useful but they do not reveal how the proposed method works internally. To be more convincing, it would be better if the paper could provide more insights on how encoding affects the final accuracy. For example, does the quality of the GCN matter when performing the encoding? If so, how do you measure the quality quantitatively? Could the existing method from Xu et al. 2019 still work by slightly adjusting its model architecture, e.g., by increasing the model depth/width or changing the hyperparameters?

Third, the evaluation on RoBERTa-base/large appears to use unoptimized baselines. For example, the RoBERTa-large baseline reaches 88.1 GLUE score on the leaderboard whereas this paper only reports 86.9. If we look at individual datasets, the RoBERTa-large results from this paper are also constantly worse than what has been reported (e.g., 90.0 vs. 90.8 for MNLI, 64.2 vs. 67.8 for CoLA). If we take the publicly reported results as the baseline, the proposed method actually performs worse than the RoBERTa-large baseline. Furthermore, it is also unclear whether the paper follows the standard practice when setting the learning rates schedules when producing the baselines for GLUE datasets. For example, RoBERTa models are often fine-tuned with warmups followed by a linear decay scheduling. However, it does not seem to be the case that the paper includes this basic baseline in the comparison. It raises concerns on whether the improvements in Table 2 are merely an effect of comparing with a set of suboptimal learning rate schedules. To be more convincing, it would be better to compare the proposed method with the best practice of fine-tuning RoBERTa-large.

Fourth, given that the proposed method requires training an additional policy network and GCN to learn the learning rate schedule, which adds additional difficulties to the already complex training of large models,  it raises concerns about the usefulness of the proposed method in practice.

---

> ### Author Response · Authors · 2021-11-16
> **Response to Reviewer TPKw (1/3)**
>
> We thank you for your valuable comments. We realize the weaknesses you have pointed out and provide our detailed response as follows.
>
> **[Q1: Novelty of the paper]** It should be emphasized that adjusting learning rate via reinforcement learning is not our primary contribution. Instead, in the paper, we proposed to fully leverage the network state information by encoding with graph neural networks, and also designed an efficient reward collection procedure to reduce computational costs. Such a GNN encoding is significantly different from previous works [1, 2]. Methods in [1, 2] only use the last layer statistics for state representation to make sure that the encoder can be generalizable to any different architectures. On the contrary, we have demonstrated that encoding the state with a graph network enables leveraging information from all the layers as well as generalizing to any different architectures. Moreover, these innovations are important as GNS significantly outperforms previous RL-based methods.
>
> **[Q2: Multiple factors can impact the final performance]** We appreciate your helpful suggestions and we do realize that there is a lack of some experiments to show the influence of each component. Here is a detailed discussion about several important factors:
> - For the first two factors you mentioned, the encoding scheme and the features, they are correlated as the features are required to be modified to integrate with the GCN encoder. In order to account for the influence of these two factors, we conducted an experiment in Section  5.6, “State representations with graph networks”, in which the input of SRLS was modified to a super vector which concatenated all layers’ states. The variant was denoted as CSS. We can observe in Table 4 that using information of all intermediate layers can slightly improve the original SRLS but the improvement was no match to the encoding scheme with graph networks in GNS. However, we notice that it is unfair to increase the dimension of input features while not modifying the model architecture and hyperparameters at the same time. We leave the discussion of larger models and hyperparameter tuning in the final paragraph.
> - As to the optimization mechanism, both GNS and SRLS from [1] adopted the same RL algorithm, PPO. It is likely that the RL method itself is suboptimal, but since the same PPO was used to train the scheduler and we did not use any alternative methods, this factor of the optimization mechanism exerts little impact on the final performance.
> - Then for the choice of hyperparameters, we used the same configuration for GNS and SRLS including the action definition, the learning rate for the actor and the critic, and the update interval. We also conducted an experiment to show that our method was not very sensitive to these hyperparameters in Appendix E.4, Sensitivity of hyperparameters.
>
> **Regarding how the quality of encoding affects the final performance:** Intuitively, the quality of encoding is expected to influence the final accuracy and the better quality leads to the better final performance. The most straightforward way to evaluate the encoding quality is to compare the performance of downstream tasks. In our paper the metric is just the performance of the target network trained with the neural scheduler. To investigate the encoding quality, we consider a list of schedulers with different encoding capabilities:
>  - SRLS
>  - SRLS + ALS~(All Layer’s States)
>  - SRLS + ALS + larger network, where the network is chosen from a 3-layer MLP with hidden size 32 (deeper), and a 2-layer MLP with hidden size 64 (wider)
>  - SRLS + ALS + larger network + tuned update interval,  $K$ is chosen from $[1, 5, 10,  20].$
>  - GNS
>  - GNS + GAT [3]
>
> (continued)

---

> > ### Author Response · Authors · 2021-11-16
> > **Response to Reviewer TPKw (2/3)**
> >
> > **[Continued Q2]** Experiments are carried out on two datasets, CIFAR10 (ResNet34) and RTE (RoBERTa-base), and results are shown in Table 14 in Appendix E. We can see that by using a super concatenated vector as the input, the performance is improved slightly, from 93.6 to 93.8 on CIFAR10 and 70.8 to 72.2 on RTE. A larger network also contributes to the performance gain to some degree but the improvement is limited, compared with the result of GNS, where the encoding scheme is significantly different from MLP networks. Note that performance with a tuned K remains the same and it indicates that our default update interval works the best. Moreover, after switching to a more powerful graph encoder, GAT, the performance can be further improved, in particular from 94.3 to 94.5 on CIFAR10.
> >
> > |  Scheduler     | CIFAR10 | RTE     |
> > | :---        |    :---     |         :---    |
> > | SRLS      | 93.6 ± 0.6       | 70.8  ± 0.7 |
> > | SRLS + ALS   | 93.8  ± 0.4       | 72.2  ± 0.5      |
> > | SRLS + ALS + larger network   | 93.9  ± 0.4     | 72.6  ± 0.4   |
> > | SRLS + ALS + larger network + tuned K  | 93.9  ± 0.4 | 72.6  ± 0.4|
> > | GNS   | 94.3  ± 0.5       | 74.4  ± 0.3     |
> > | GNS + GAT   | **94.5  ± 0.4**        | **74.7 ± 0.3**       |
> >
> > To sum up, leveraging more information (SRLS -> SRLS + ALS) does enhance the scheduler. Using a better network architecture (a larger  network) and hyperparameter configuration also helps but encoding with merely MLPs is not sufficient to make full use of all intermediate layers’ information. By representing the target network as a directed  graph and adopting an encoding scheme of GCN, the encoder quality improves and the corresponding scheduler is much better. With a stronger encoder GAT, the final performance can be further improved, which validates that the encoding quality matters for the learned scheduler.
> >
> > **[Q3: ImageNet results]** To make our method more convincing, we have added the performance of different schedulers on ImageNet in Table 3 of transferred performance of GNS. Still, we adopt the best hyperparameter configuration from ResNet34 on CIFAR10 to train the same model structure on ImageNet. For GNS,  we just apply the policy network trained under CIFAR10 to adjust learning rate for training on ImageNet. It can be seen that our GNS works consistently well and leads to the best test accuracy. In particular, transferred performance of GNS is 73.1% and is better than the best baseline of 71.6%.
> >
> > |  Scheduler | CIFAR10 -> ImageNet |
> > | :---       |    :----:           |
> > | Constant      | 68.9       |
> > | Polynomial   | 70.3       |
> > | Cosine   | 70.5     |
> > | Hypergraident  | 70.0 |
> > | SRLS   | 71.6       |
> > | GNS   | **73.1**   |
> >
> > **[Q4: Evaluation on GLUE datasets]** Thanks for pointing out the inconsistency between our results and public ones. In fact, the public results are obtained from ensembles of 5-7 single-task models with different initializations, while in our paper we focus on the performance of a single-task model. In the original Roberta paper [4], only the results of the single model on the development set are reported. Thus, we add the dev result for a fair comparison in Table 15 in Appendix E. It can be observed that GNS performs consistently better on the dev set as well. On the other hand, we admit that the standard practice for fine-tuning on GLUE datasets is a warmup procedure followed by a linear decay scheduling. We initially adopted this mechanism as a baseline in our experiments but it turned out that the linear decay performed similarly to the polynomial one, as the former can be regarded as a special case of the latter. To make our results more convincing, we modify the table to include the reported results in the paper as well as the results from the standard practice. From Table 15, we can tell that our proposed GNS (89.6 on average) outperforms this standard method and the reported results in [4] (88.9 on average), indicating that the improvements are not due to the suboptimal learning rate schedules, but from our better learned policy.
> >
> > |  Scheduler | CoLA | SST-2 | MRPC | QQP | STS-B | MNLI | QNLI | RTE | Avg.
> > |  :----: | :----:| :----:| :----:| :----:| :----:| :----:| :----:| :----:| :----:|
> > | Constant      | 68.8 | 96.1 | 90.3| 92.0| 92.1| 90.0| 94.4| 86.2| 88.7|
> > | Polynomial   | 69.1|96.3|90.6|92.2|92.4|90.2|94.8|86.5|89.0|
> > | Cosine   | 69.0 | 96.4| 90.5 | 92.2 | 92.3| 90.1| 94.6| 86.7| 89.0|
> > | Hypergraident  | 68.6 | 96.3 | 90.5| 92.0| 92.1| 90.1| 94.5| 86.5| 88.8|
> > | SRLS   | 67.8 | 95.9 | 89.7 | 91.6 | 92.1 | 89.9 | 94.2| 86.0| 88.4|
> > | GNS   | **70.5** | **97.1** | **91.2** | **92.7** | **92.9** | **90.5** | **94.9** | **87.4** | **89.6** |
> > | [4] | 68.0 | 96.4| 90.9| 92.2| 92.4 | 90.2 | 94.7| 86.6 | 88.9|
> > | Standard   | 68.8 | 96.3 | 90.7 | 92.1 | 92.3 | 90.2 | 94.7 | 86.5 | 88.9|
> >
> > (continued)

---

> > > ### Author Response · Authors · 2021-11-16
> > > **Response to Reviewer TPKw (3/3)**
> > >
> > > **[Q5: Practical usefulness of the method]** Even without learning the policy network, we still need to fine-tune the hyperparameters like learning rate scheduling to reach a good result. In this paper, we have shown that training GNS can be more efficient than the commonly-used grid search method, and GNS can achieve better performance at the same time. We also compare GNS to a more efficient and effective HPO method, CMA-ES [5], and as shown in Figure 5 in Appendix E, it is advantageous over CMA-ES as well. Furthermore, GNS can be generalized to different datasets and networks directly or with slight fine-tuning, which makes it promising to be leveraged in real-world applications.
> > >
> > > [1] Xu, Zhen, et al. "Learning an adaptive learning rate schedule." arXiv preprint arXiv:1909.09712 (2019).
> > >
> > > [2] Xu, Chang, et al. "Reinforcement learning for learning rate control." arXiv preprint arXiv:1705.11159 (2017).
> > >
> > > [3] Veličković, Petar, et al. "Graph attention networks." arXiv preprint arXiv:1710.10903 (2017).
> > >
> > > [4] Liu, Yinhan, et al. "Roberta: A robustly optimized bert pretraining approach." arXiv preprint arXiv:1907.11692 (2019).
> > >
> > > [5] Hansen, Nikolaus. "The CMA evolution strategy: A tutorial." arXiv preprint arXiv:1604.00772 (2016).

---

> > > > ### Comment · Reviewer_TPKw · 2021-11-30
> > > > **Reply to rebuttal**
> > > >
> > > > Most of the authors' replies are satisfactory, and I appreciate the effort in adding experiments. The final version of the paper would definitely benefit from these clarification and experiment results. I increased my score to reflect my position.

---

> ### Author Response · Authors · 2021-11-28
> **Follow-up comments from Reviewer TPKw**
>
> Dear Reviewer TPKw,
>
> We sincerely hope to have further discussion with you regarding your concerns and our responses. We are happy to answer any additional questions and provide more information.
>
> We would greatly appreciate it if you could reply to our response. Thank you!
>
> Best,
>
> Paper124 Authors

---

### Author Response · Authors · 2021-11-19
**Follow-up questions**

Dear reviewers,

Since the first discussion period will end soon, we are just wondering whether there are any updates or follow-up questions about our paper so that we can have enough time to respond and update the draft. Your suggestions and comments are valuable to us.

Thanks again for your detailed and constructive reviews.

Best,

Paper124 Authors

---

### Author Response · Authors · 2021-11-26
**Revision summary**

We thank all reviewers for the careful and constructive reviews. We have revised the paper accordingly and summarized changes as follows:
- We evaluated the performance of GNS on a large-scale dataset, ImageNet and included the results in Table 3. It can be seen that GNS performed consistently better than other schedulers on this large dataset.
- We investigated the impact of multiple factors that might affect the performance of the learned scheduler, including the feature form, the network width/depth, hyperparameters, and the encoder network structure. Detailed results are presented in Appendix E.5, which have validated the effectiveness of representing the state as a graph.
- We included the original results from RoBERTa for a fair comparison in Appendix E.6 and showed that the improvements are not due to the suboptimal learning rate schedules, but from our better learned policy
- We compared our searching procedure of GNS with a more effective hyperparameter optimization algorithm in Appendix E.7, CMA-ES, and observed that GNS was as efficient as CMA-ES and could find a better learning rate scheduling policy.
- We conducted an ablation study of the choice of raw features in Appendix E.8, and demonstrated that these components of raw features contributed to the training of GNS.
- We added a comparison with different forms of action in Appendix E.9 and showed that our current action(scaling by a factor) worked most stably and effectively.
- We evaluated GNS on a simple MLP network in Appendix E.10 and observed that GNS still outperformed all the baselines.
- We included training loss curves on Fashion MNIST and CIFAR10 in Appendix E.11 to better show the advantages of GNS.

We hope this summary can help you better understand the paper.

Best,

Paper124 Authors

---

### Decision · Program_Chairs · 2022-01-20

**Decision:**

Accept (Poster)

**Comment:**

### Description
The paper develops a new automatic scheduler to schedule the learning rate during the training. The scheduler has access to the current training state summarized by certain statistics of weights and gradients in all layers and the loss history. It is trained by reinforcement learning with the reward derived from the progress with respect to the performance measure, such as validation accuracy. The key innovations are the design of the state vector using graph convolutional neural networks and empirical improvements to the reward function. The main claim is that GCNs allow to take into account the architecture of the network to be trained and the state of all layers, which, authors hypothesize and demonstrate experimentally, improves performance and transferability of the scheduler across networks and tasks.

### Decision
The reviewers recommendations after the rebuttal settled on 4 x "marginally above" and one "accept". Respectively, I recommend to accept. I recommend a poster based on the reception by reviewers: the novelty was assessed as limited because the idea of an automatic schedulers based on RL with a similar learning strategy belongs to the prior work. On the strong side, the paper satisfied all requests by reviewers for the experiments, regarding alternative methods, large datasets and ablation studies demonstrating that it is indeed the new architecture that allows to achieve a significant improvement, making it a solid improvement step. Amongst alternative methods the paper considers all viable alternatives: a function-based schedulers, a hyper-gradient method, and a the RL based scheduler, optimized in hyperparameters.

### Discussion
There was no significant non-public discussion. As an additional feedback, let me just share my observations.

What is somewhat unclear in that the paper starts by discussing the directed graph of a feed-forward network, then it proposes to run GCN on it, which is undirected. Then the hierarchical method is proposed, which runs GCN on each block sequentially while taking the aggregated input from the preceding block. This makes it a directed processing method on the level of block. I wonder whether the directed processing is desired or not desired here? Can a sequential processing summarize the network state efficiently on its own, similar to feedforward propagation, without the global averaging proposed?

It was not very clear to me from the paper what is the meaning of the batch size for training the GSN, and respectively what the batch normalization is doing there.

From some 100 mile perspective, it seems to me that whatever efficient optimization can be performed on the validation set, it helps. It does not matter so much what is varied: the learning rate schedule, other hyperparameters or even the network architecture. So in a sense it is not surprising that one can improve. What is more interesting is that the learned schedulers are generalizable / transferable, as demonstrated in Section 5.4 (changing the architecture or going from CIFAR to ImageNet while keeping the scheduler).

The work has done quite a lot on the experimental side with the baseline GCN model that they proposed. It seems to have still lots of potential via different possible enhancements. For example, what authors mentioned, including exponentially weighted running averages of gradients and squared gradients into the features. They already tried GAT instead of GCN as proposed by reviewers. There may be hyperparameters other than those controlling the learning rate. Some such hyperparameters, e.g. momentum, are apparently tightly coupled with the learning rate. The paper does not discuss how to tune them together with GNS. It could be a difficulty. On the other hand, they can be potentially scheduled with the same GNS.

When considering SGD with momentum (which is not used in the paper), please note that the common use of a momentum parameter $\mu$ actually mixes the learning rate together with the smoothing parameter controlling the exponentially weighted averaging. So if one wants to control the learning rate alone, it is better to implement the gradient smoothing is done in .e.g. Adam, with its hyperparameter $\beta_1$.

---

> ### Public Comment · ~Yuanhao_Xiong1 · 2022-02-09
> **Reply to program chairs**
>
> Thanks for your detailed comments. Here we made some clarifications on your questions.
>
> For the hierarchical processing, it is suggested in [1] and can keep the directed property of the original network structure and scales  naturally with the design of repeated modules. We also evaluated using one GCN for the whole graph but the performance was not as good as the current design. As to the global averaging operation, it is the aggregation operation to summarize the whole graph and is required for the subsequent action prediction.
>
> The batch size is for the training of the actor and the critic in reinforcement learning. During the inference phase, i.e., predicting the action, we just used the evaluation mode with a fixed batch normalization layer.
>
> All your other suggestions are valuable to us and adding more informative features as well as considering other hyperparameters like momentum can be potential future directions. We sincerely appreciate your effort in our paper.
>
> [1] Zhang, Chris, Mengye Ren, and Raquel Urtasun. "Graph hypernetworks for neural architecture search." arXiv preprint arXiv:1810.05749 (2018).